# Out-of-Distribution Detection by Leveraging Between-Layer Transformation Smoothness

**Fran Jelenić**[1,2]   **Josip Jukić**[1,2]   **Martin Tutek**[3]   **Mate Puljiz**[2]   **Jan Šnajder**[1,2]

[1]TakeLab, [2]Faculty of Electrical Engineering and Computing, University of Zagreb, Croatia
[3]UKP Lab, Technical University of Darmstadt, Germany
`{fran.jelenic, josip.jukic, mate.puljiz, jan.snajder}@fer.hr`
`tutek@ukp.informatik.tu-darmstadt.de`

## Abstract

Effective out-of-distribution (OOD) detection is crucial for reliable machine learning models, yet most current methods are limited in practical use due to requirements like access to training data or intervention in training. We present a novel method for detecting OOD data in Transformers based on transformation smoothness between intermediate layers of a network (BLOOD), which is applicable to pre-trained models without access to training data. BLOOD utilizes the tendency of between-layer representation transformations of in-distribution (ID) data to be smoother than the corresponding transformations of OOD data, a property that we also demonstrate empirically. We evaluate BLOOD on several text classification tasks with Transformer networks and demonstrate that it outperforms methods with comparable resource requirements. Our analysis also suggests that when learning simpler tasks, OOD data transformations maintain their original sharpness, whereas sharpness increases with more complex tasks.

## 1 Introduction

Machine learning (ML) models' success rests on the assumption that the model will be evaluated on data that comes from the same distribution as the data on which it was trained, the *in-distribution* (ID) data. However, models deployed in noisy and imperfect real-world scenarios often face data that comes from a different distribution, the *out-of-distribution* (OOD) data, which can hinder the models' performance. The task of discerning between ID and OOD data is commonly referred to as *OOD detection* (Yang et al., 2021).

Owing to their consistent state-of-the-art performance across diverse ML tasks (Abiodun et al., 2018), Deep Neural Networks (DNNs) have garnered significant attention in OOD detection research. While popular baselines make use of the model's posterior class probabilities (Hendrycks & Gimpel, 2017), the issue of overconfidence in DNNs (Guo et al., 2017) frequently erodes the credibility of these probabilities. An alternative is offered by the group of methods that leverage the fundamental concept of DNNs, namely, representation learning. Because a DNN encodes similar instances closely in its representation space, an OOD instance can be identified based on the distance between its representation and the representations of other instances in the training set (Lee et al., 2018). The downside of these methods, however, is that they require the presence of training data during prediction or involve intervention in the model's training procedure. This is a significant practical limitation, as using third-party models pre-trained on non-public data is increasingly the standard practice. A case in point is the Hugging Face Transformers library (Wolf et al., 2020), which provides community models but often lacks comprehensive details about their training.

An obvious way to close the resource gap is to rely on OOD detection methods with minimal prerequisites. However, current OOD detection research has largely ignored the differing prerequisites among OOD detection methods, often leading to comparisons that treat methods with varying prerequisites equally, disregarding the question of practical applicability. From a practical perspective,

it makes sense to group OOD detection methods into the following three categories:[1] (1) *Black-box*, for methods capable of operating on black-box models (i.e., having access only to input-output mappings) and thus suitable for models integrated into a product; (2) *White-box*, for methods that require access to the model's weights and have knowledge about its architecture, and are thus readily applicable to third-party pre-trained models; and (3) *Open-box*, for methods with unrestricted access to model and training resources, allowing for interventions in the training process and/or access to training data or separate OOD train or validation sets.

In this paper, we focus on the OOD detection for the Transformer architecture (Vaswani et al., 2017), which has emerged as the predominant architecture in numerous ML domains. We introduce a novel OOD detection method that leverages the inherent differences in how Transformers process ID and OOD data. The method is white-box and has the potential for broad practical applicability. More concretely, our **B**etween **L**ayer **O**ut-**O**f-**D**istribution (BLOOD) Detection method estimates the smoothness of between-layer transformations of intermediate representation, building on the insight that these transformations tend to be smoother for ID data than for OOD data. We evaluate BLOOD on Transformer-based pre-trained large language models applied to text classification, the most prevalent task in natural language processing (NLP), and find that it outperforms other state-of-the-art OOD detection white-box methods and even some open-box methods. We further analyze BLOOD to probe into the underlying causes of the differences between how ID and OOD intermediate representations are transformed and evaluate BLOOD on two other types of distribution shifts – semantic and background shift. We provide code and data for our experiments.[2]

The contributions of this paper are as follows: **(1)** We propose BLOOD, a novel method for OOD detection applicable even in cases when only the model's weights are available, e.g., third-party pre-trained models which are becoming *de facto* standard in many fields. BLOOD uses the information about the smoothness of the between-layer transformations of intermediate representations. We quantify this smoothness using the square of the Frobenius norm of the Jacobian matrix, for which we provide an unbiased estimator to alleviate computational limitations. **(2)** Our experiments on Transformer-based pre-trained large language models for the task of text classification show that BLOOD outperforms other state-of-the-art white-box OOD detection methods. Additionally, our results indicate that the performance advantages are more prominent when applied to complex datasets as opposed to simpler ones. We also show that BLOOD is more effective in detecting background shift than semantic shift. **(3)** Following our main insight that between-layer representation transformations of ID data tend to be smoother from that of OOD data, we analyze the source of this difference. We find that the learning algorithm is more focused on changing the ID region of intermediate representation space, smoothing the between-layer transformations of ID data in the process. At the same time, the OOD region of the intermediate representation space is largely left unchanged, except in some scenarios, e.g., for more complex tasks, when the OOD region of the space is also changed and sharpened as a consequence.

## 2    RELATED WORK

OOD detection methods are typically categorized based on their underlying mechanism, for example, into output-based, gradient-based, distance-based, density-based, and Bayesian methods (Yang et al., 2021). Another, and arguably more practically relevant, categorization would factor in the necessary prerequisites for these methods, distinguishing between black-box, white-box, and open-box methods as introduced earlier. In the following, we provide a brief overview of the most prominent OOD detection methods through this lens.

**Black-box.** Methods with minimal prerequisites typically rely on posterior class probabilities, assuming that when a model is uncertain about an instance, the instance is more likely to be OOD. A commonly used baseline quantifies the uncertainty of an instance as the negative of the model's maximum softmax probability for that instance (Lee et al., 2018). A straightforward modification employs the entropy of softmax probabilities rather than the maximum value. Liu et al. (2020b) proposed using energy scores instead of softmax scores to overcome the issue of DNN overconfidence.

---

[1]Gomes et al. (2022) employ similar terminology to refer to which parts of the model one can access (e.g., its outputs, inputs, or intermediate representations). In contrast, we use these terms to characterize the resources an OOD detection method requires.

[2]`https://github.com/fjelenic/between-layer-ood`

**White-box.** Gal & Ghahramani (2016) proposed using Monte-Carlo dropout to more reliably estimate the model's uncertainty, showing that dropout (Srivastava et al., 2014) with DNNs approximates Bayesian inference. Although Monte-Carlo dropout outperforms vanilla posterior probabilities in OOD detection (Ovadia et al., 2019), it is computationally expensive as it requires multiple forward passes. Another way of leveraging the access to model's architecture is to use gradients to implicitly measure the uncertainty of the model's predictions (Oberdiek et al., 2018; Huang et al., 2021). Gradient methods primarily employ the gradient norm to gauge the difference between the model's posterior distribution and the ideal distribution. Djurisic et al. (2023) detect OOD data by pruning and adjusting the representations of the model, grounded in the intuition that the representations generated by contemporary DNNs tend to be excessive for their designated tasks.

**Open-box.** Because DNNs posterior probabilities tend to exhibit overconfidence, Guo et al. (2017) suggested using temperature scaling to calibrate the model's posterior probabilities, which entails the usage of a separate validation set. To get higher quality predictive uncertainty estimates, Lakshminarayanan et al. (2017) train an ensemble of differently initialized models and combine their predictions. Although ensembles are robust to different distributional shifts (Ovadia et al., 2019), they impose a significant computational and memory overhead because they require training and keeping in memory of multiple models. Agarwal et al. (2022) extend the gradient-based methods by leveraging the variance of the gradient of the predicted label w.r.t. the input through different checkpoints during training. A popular approach to OOD detection for DNNs revolves around the utilization of information related to distances in the representation space (Lee et al., 2018; Van Amersfoort et al., 2020; Liu et al., 2020a; Hsu et al., 2020; Kuan & Mueller, 2022; Sun et al., 2022). However, these approaches require access to the training data or changes in the standard training procedure. Yet another set of methods relies on exposing the model to OOD samples during training to improve the performance on OOD detection task (Hendrycks et al., 2019; Thulasidasan et al., 2021; Roy et al., 2022). Still, a major practical limitation of these methods is the necessity for OOD data, whose entire distribution is typically unknown in real-world scenarios. Several post-hoc methods also need OOD data, but for validation sets to optimize their method's hyperparameters (Liang et al., 2018; Sun et al., 2021; Sun & Li, 2022).

## 3 PRELIMINARIES

### 3.1 PROBLEM STATEMENT

Let instance $\boldsymbol{x} \in \mathbb{R}^d$ be a $d$-dimensional feature vector and $y \in \{0, \ldots, C-1\}$ be its corresponding class in a $C$-way classification task. We train a classifier on the dataset $\mathcal{D} = \{(\boldsymbol{x}_n, y_n)\}_{n=1}^N$ consisting of $N$ instances i.i.d. sampled from the distribution $p(\boldsymbol{x}, y)$. The objective of the learning algorithm is to model the conditional distribution $p(y|\boldsymbol{x})$ based on $\mathcal{D}$ by estimating the parameters $\boldsymbol{\theta}$ of the distribution $p_{\boldsymbol{\theta}}(y|\boldsymbol{x})$ that is as close as possible to the true conditional distribution.

The goal of an OOD detection method is to determine the uncertainty score $\mathcal{U}_{\boldsymbol{x}} \in \mathbb{R}$ of an instance $\boldsymbol{x}$, such that there exist $\epsilon \in \mathbb{R}$ for which both $\mathbb{P}_{\boldsymbol{x} \sim p(\boldsymbol{x},y)}(\mathcal{U}_{\boldsymbol{x}} < \epsilon)$ and $\mathbb{P}_{\boldsymbol{x} \sim q(\boldsymbol{x},y)}(\mathcal{U}_{\boldsymbol{x}} > \epsilon)$ are close to unity whenever $q(\boldsymbol{x}, y)$ is a distribution sufficiently different from $p(\boldsymbol{x}, y)$. In practice, there can never exist a scoring function that perfectly discriminates between ID examples (generated by $p(\boldsymbol{x}, y)$) and OOD examples (generated by $q(\boldsymbol{x}, y)$). Nevertheless, even reasonable attempts can prove valuable in real-world scenarios.

### 3.2 INTUITION

Transformers work by mapping the input features onto a high-dimensional representation space through $L$ layers using the self-attention mechanism, creating a representation of the data suitable for the task at hand. The mapping is realized as a composition of several attention layers, where each layer creates an intermediate representation of the input. It has been show that Transformer-based models tend to gradually progress from input features towards more abstract representation levels through layers, i.e., lower layers model lower-level features, while upper layers model higher-level features. For example, Peters et al. (2018); Tenney et al. (2019); Jawahar et al. (2019) showed that large Transformer-based language models create text representations that progress gradually from representations that encode morphological and syntactic information at the lower layers to representations that encode semantic meaning in the upper layers. Likewise, Vision Transformers

(ViT) (Dosovitskiy et al., 2021), which are garnering popularity in computer vision, were shown to process images in a similar fashion (Ghiasi et al., 2022).

We hypothesize that during the model's training, the model learns smooth transformations between layers corresponding to natural and meaningful progressions between abstractions for ID data. We further hypothesize that these progressions will not match OOD data, hence the transformations will not be smooth for OOD data. Thus, if we could measure the smoothness of transformations in representations between layers, we could in principle differentiate between ID and OOD data. We also speculate that the difference in smoothness of transformations between ID and OOD data should be emphasized in the upper layers of a Transformer. Lower layers typically represent low-level features that are more universal, whereas upper layers tend to cluster instances around task-specific features that are not shared between ID and OOD data, potentially creating a mismatch in levels of abstraction.

### 3.3 Our method

Assume an $L$-layered deep neural network $\boldsymbol{f} : \mathbb{R}^{d_0} \to [0, 1]^C$ was trained to predict the probabilities of $C$ classes for a $d_0$-dimensional input $\boldsymbol{x}$. Let $\boldsymbol{f}$ be a composition of $L$ intermediate functions, $\boldsymbol{f}_L \circ \cdots \circ \boldsymbol{f}_l \circ \cdots \circ \boldsymbol{f}_1$, where $\boldsymbol{f}_l : \mathbb{R}^{d_{l-1}} \to \mathbb{R}^{d_l}$, $l = 1, \ldots, L-1$, correspond to intermediate network layers, while $\boldsymbol{f}_L$ corresponds to the last layer, mapping to a vector of logits to which softmax function is applied to obtain the conditional class probabilities. We denote the intermediate representation of $\boldsymbol{x}$ in layer $l$ as $\boldsymbol{h}_l$, defined as $\boldsymbol{h}_l = (\boldsymbol{f}_l \circ \cdots \circ \boldsymbol{f}_1)(\boldsymbol{x})$.

We now need to quantify how smoothly an intermediate representation is transformed from layer $l$ to layer $l + 1$. To this end, we first need to define what we consider a smooth transformation. We say a representation $\boldsymbol{h}_l$ is transformed smoothly if there is not a large difference in how it is mapped from layer $l$ onto layer $l + 1$ compared to how its infinitesimally close neighborhood is mapped.

Let $\phi_l(\boldsymbol{x})$ be the degree of smoothness of the transformation between representation $\boldsymbol{h}_l$ and representation $\boldsymbol{h}_{l+1}$ for input $\boldsymbol{x}$. To calculate $\phi_l(\boldsymbol{x})$, we compute the Jacobian matrix $\frac{\partial \boldsymbol{f}_{l+1}}{\partial \boldsymbol{h}_l} = \boldsymbol{J}_l :$ $\mathbb{R}^{d_l} \to \mathbb{R}^{d_{l+1} \times d_l}$, and take the square of its Frobenius norm:

$$\phi_l(\boldsymbol{x}) = \|\boldsymbol{J}_l(\boldsymbol{h}_l)\|_F^2 = \sum_{i=1}^{d_{l+1}} \sum_{j=1}^{d_l} \left( \frac{\partial (f_{l+1})_i}{\partial (h_l)_j} \right)^2 \tag{1}$$

In the most popular ML libraries, gradients of a function are computed through automatic differentiation (AD), which comprises both forward mode and backward mode. Forward mode AD computes the values of the function and a Jacobian-vector product. Computing the full Jacobian matrix $\boldsymbol{J}(\boldsymbol{x})$ with AD is computationally expensive as it requires $d$ forward evaluations of $\boldsymbol{J}(\boldsymbol{x})\boldsymbol{e}^{(i)}$, $i = 1, \ldots, d$, where $\boldsymbol{e}^{(i)}$ are standard basis vectors, computing the Jacobian matrix one column at a time. In the case of modern DNNs with high-dimensional hidden layers, computing full Jacobians could render our method unfeasible. To reduce computational complexity, we derive an unbiased estimator of $\phi_l(\boldsymbol{x})$ by leveraging Jacobian-vector product computation through forward mode AD.

**Corollary 1.** *Let $\boldsymbol{J}(\boldsymbol{x}) \in \mathbb{R}^{m \times n}$ be a Jacobian matrix, and let $\mathbf{v} \in \mathbb{R}^n$ and $\mathbf{w} \in \mathbb{R}^m$ be random vectors whose elements are independent random variables with zero mean and unit variance. Then, $\mathbb{E}[(\mathbf{w}^\intercal \boldsymbol{J}(\boldsymbol{x})\mathbf{v})^2] = \|\boldsymbol{J}(\boldsymbol{x})\|_F^2$.*

We prove Corollary 1 in the Appendix B by providing a proof for more general Theorem 1. As for the intuition behind the corollary, the Jacobian-vector product $\boldsymbol{J}(\boldsymbol{x})\mathbf{v}$ gives us an appropriately scaled gradient with respect to the change of the input in the direction of vector $\mathbf{v}$. Further multiplying the Jacobian-vector product $\boldsymbol{J}(\boldsymbol{x})\mathbf{v}$ by the random vector $\mathbf{w}$ from the left projects the calculated directional gradient $\boldsymbol{J}(\boldsymbol{x})\mathbf{v}$ on the vector $\mathbf{w}$, i.e., it quantifies the extent to which the output changes in the direction of $\mathbf{w}$ when the input changes in the direction of $\mathbf{v}$. Squaring the vector-Jacobian-vector product then gives an estimate of the sum of squared entries of the Jacobian, i.e., the square of its Frobenius norm. Squaring also handles negative values (in cases when the angle between the directional gradient $\boldsymbol{J}(\boldsymbol{x})\mathbf{v}$ and the vector $\mathbf{w}$ is obtuse), since we are interested in the overall smoothness as defined by Frobenius norm rather than the direction of the specific gradient.[3]

---

[3]Our notion of smoothness extends from Lipschitz continuity, where the spectral norm of the Jacobian acts as a lower bound for the Lipschitz constant (Rosca et al., 2020). Since all matrix norms are equivalent, we use the Frobenius norm, which can be efficiently computed, rather than the spectral norm to capture smoothness.

To calculate the unbiased estimate $\hat{\phi}_l(\boldsymbol{x})$ of $\phi_l(\boldsymbol{x})$, we use a sample of $M$ pairs of random vectors $\mathbf{v}_l \sim \mathcal{N}(\mathbf{0}_n, \boldsymbol{I}_n)$ and $\mathbf{w}_l \sim \mathcal{N}(\mathbf{0}_m, \boldsymbol{I}_m)$, and define $\hat{\phi}_l(\boldsymbol{x})$ as:

$$\hat{\phi}_l(\boldsymbol{x}) = \frac{1}{M} \sum_{i=1}^{M} \left( \mathbf{w}_{l,i}^{\mathsf{T}} \boldsymbol{J}_l(\boldsymbol{h}_l) \mathbf{v}_{l,i} \right)^2 \tag{2}$$

BLOOD uses $\hat{\phi}_l(\boldsymbol{x})$ as the uncertainty score of an instance $\boldsymbol{x}$. In our experiments, we consider two variations of BLOOD: (1) the average of scores for all layers $\text{BLOOD}_M = \frac{1}{L-1} \sum_{l=1}^{L-1} \hat{\phi}_l(\boldsymbol{x})$, and (2) the score for the projection of $\text{BLOOD}_L = \hat{\phi}_{L-1}(\boldsymbol{x})$. We use the two variants to assess the impact of layer choice, as we hypothesize that BLOOD will perform better on upper layers, given that lower layers capture low-level, general features.

## 4 EXPERIMENTS

### 4.1 EXPERIMENTAL SETUP

We evaluate BLOOD on several text classification datasets using two transformer-based (Vaswani et al., 2017) large pre-trained language models, RoBERTa (Liu et al., 2019) and ELECTRA (Clark et al., 2020), known for their state-of-the-art performance across a wide range of NLP tasks. We calculate the BLOOD score using samples of size $M = 50$ to estimate $\hat{\phi}_l(\boldsymbol{x})$ of `[CLS]` token's representations between layers. We use eight text classification datasets for ID data: SST-2 (**SST**; Socher et al., 2013), Subjectivity (**SUBJ**; Pang & Lee, 2004), AG-News (**AGN**; Zhang et al., 2015), and TREC (**TREC**; Li & Roth, 2002), BigPatent (**BP**; Sharma et al., 2019), AmazonReviews (**AR**; McAuley et al., 2015), MovieGenre (**MG**; Maas et al., 2011), 20NewsGroups (**NG**; Lang, 1995). We use One Billion Word Benchmark (**OBW**) (Chelba et al., 2014) for OOD data, similarly to Ovadia et al. (2019), because of the diversity of the corpus. We subsample OOD datasets to be of the same size as their ID test set counterparts. Appendix C provides more details about the models, datasets, and training procedures.

We compare BLOOD to several state-of-the-art black-box and white-box OOD detection methods: (1) **Maximum softmax probability** (**MSP**) – the negative posterior class probability of the most probable class, $-\max_c p(y = c|\boldsymbol{x})$, often considered a baseline OOD detection method (Hendrycks & Gimpel, 2017); (2) **Entropy** (**ENT**) – the entropy of the posterior class distribution, $\mathbb{H}[Y|\boldsymbol{x}, \boldsymbol{w}]$; (3) **Energy** (**EGY**) – a density-based method that overcomes the overconfidence issue by calculating energy scores from logits $-\log \sum_{i=0}^{C-1} e^{f_L(\boldsymbol{x})_i}$ instead of softmax scores (Liu et al., 2020b); (4) **Monte-Carlo dropout** (**MC**) – the entropy of predictive distribution obtained using Monte–Carlo dropout (Gal & Ghahramani, 2016). We use $M = 30$ stochastic forward passes to estimate uncertainty; (5) **Gradient norm** (**GRAD**) – the L2-norm of the penultimate layer's gradient of the loss function with most likely class considered as a true class (Oberdiek et al., 2018). (6) **Activation shaping** (**ASH**) – removing 90% of the smallest activations and adjusting the rest using ASH-S method in the penultimate layer (Djurisic et al., 2023).

Additionally, we compare BLOOD to three standard open-box OOD detection methods. Given that these methods entail considerably more prerequisites compared to BLOOD and other white/black-box methods, this comparison is intended solely as a reference point: (1) **Rectified Activations** (**ReAct**) – setting the values of the activations in the penultimate layer to be at most the 90th percentile of the activations of the training data (Sun et al., 2021). (2) **Ensemble** (**ENSM**) – an ensemble of $M = 5$ models of the same type, e.g., an ensemble of five RoBERTa or ensemble of five ELECTRA models, (Lakshminarayanan et al., 2017); (3) **Temperature scaling** (**TEMP**) – introduces a temperature parameter $T$ into the softmax function such that it minimizes the negative log-likelihood on the ID validation set (Guo et al., 2017); (4) **Mahalanobis distance** (**MD**) – Mahalanobis distance of a query instance in the representation space with respect to the closest class-conditional Gaussian distribution (Lee et al., 2018).

### 4.2 OOD DETECTION PERFORMANCE

As the performance measure for OOD detection, we follow the standard practice and use the area under the receiver operating characteristic curve (AUROC) metric (in Appendix H, we report the

Table 1: The performance of OOD detection methods measured by AUROC (%). The best-performing white/black-box method is in **bold**. Open-box methods that outperform the best-performing white/black-box method are in **bold**. Higher is better. We test the performance of $\text{BLOOD}_M$ and $\text{BLOOD}_L$ against the MSP baseline using the one-sided Man-Whitney U test; significant improvements ($p < .05$) are indicated with asterisks ($^*$).

| Model | Dataset | White-box/Black-box | | | | | | | | Open-box | | | |
|---|---|---|---|---|---|---|---|---|---|---|---|---|---|
| | | $\text{BLOOD}_M$ | $\text{BLOOD}_L$ | MSP | ENT | EGY | MC | GRAD | ASH | ReAct | ENSM | TEMP | MD |
| RoBERTa | SST | 50.56 | **72.83** | 71.69 | 71.69 | 71.61 | 68.28 | 71.76 | 67.22 | 69.55 | 69.03 | 71.64 | **85.36** |
| | SUBJ | 52.02 | 74.66 | 74.55 | 74.55 | 75.79 | 74.21 | 74.93 | **79.27** | 73.33 | 76.68 | 74.41 | **93.47** |
| | AGN | 77.46 | 61.95 | 73.57 | 73.80 | 76.36 | **77.55** | 73.58 | 72.54 | 77.10 | **80.35** | 75.38 | **82.63** |
| | TREC | 69.63 | 95.30 | 96.20 | **96.40** | 96.28 | 95.68 | 96.14 | 90.36 | 96.05 | **96.87** | **96.74** | **96.74** |
| | BP | 87.20$^*$ | **89.53**$^*$ | 70.15 | 72.82 | 85.84 | 74.29 | 73.11 | 82.18 | 86.19 | 79.39 | 86.01 | **97.35** |
| | AR | 91.41$^*$ | **93.20**$^*$ | 89.06 | 89.96 | 92.39 | 90.59 | 88.65 | 91.42 | 92.65 | 92.44 | 92.25 | **98.35** |
| | MG | **88.15**$^*$ | 85.23$^*$ | 75.02 | 76.60 | 86.45 | 79.98 | 74.28 | 81.62 | 87.30 | 76.98 | 84.30 | **95.12** |
| | NG | **83.53**$^*$ | 72.02 | 77.49 | 78.76 | 82.65 | 79.32 | 76.93 | 77.73 | 83.17 | 80.77 | 82.87 | **90.68** |
| ELECTRA | SST | 74.36 | **78.11**$^*$ | 73.84 | 73.84 | 71.97 | 70.81 | 73.82 | 67.92 | 71.18 | 73.81 | 73.58 | **78.85** |
| | SUBJ | 74.10 | 77.41 | **78.17** | **78.17** | 70.46 | 77.71 | 78.11 | 75.11 | 68.33 | **79.23** | 78.20 | **81.59** |
| | AGN | 65.67 | **80.28** | 76.80 | 77.01 | 79.75 | 79.55 | 76.57 | 77.96 | 79.46 | 79.50 | 78.31 | **86.10** |
| | TREC | 97.48 | **98.90**$^*$ | 97.26 | 97.56 | 97.48 | 96.21 | 97.07 | 90.18 | 97.50 | 97.55 | 98.20 | 97.54 |
| | BP | 86.06$^*$ | **96.72**$^*$ | 78.56 | 81.75 | 84.63 | 83.04 | 76.77 | 79.81 | 85.26 | 84.20 | 84.69 | **98.28** |
| | AR | 84.58 | **91.66**$^*$ | 87.74 | 88.44 | 90.64 | 88.53 | 87.52 | 83.96 | 91.01 | **91.98** | 90.35 | **95.47** |
| | MG | 80.52 | **90.63**$^*$ | 73.83 | 74.78 | 80.41 | 76.67 | 73.35 | 71.84 | 81.22 | 76.86 | 78.47 | **92.96** |
| | NG | 77.61 | **82.47**$^*$ | 76.45 | 77.73 | 80.83 | 79.11 | 75.97 | 74.50 | 80.95 | 79.93 | 80.75 | **89.13** |

results using two other commonly used metrics, AUPR-IN and FPR@95TPR; these gave qualitatively identical results as AUROC). The OOD detection task is essentially a binary classification task, with AUC corresponding to the probability that a randomly chosen OOD instance will have a higher uncertainty score than a randomly chosen ID instance (Fawcett, 2006). The AUROC for random value assignment is 50%, while a perfect method achieves 100%. We run each experiment five times with different random seeds and report the mean AUROC.

OOD detection performance is shown in Table 1. The first observation is that BLOOD outperforms other white/black-box methods. Secondly, $\text{BLOOD}_L$ outperforms other white/black-box methods more often than $\text{BLOOD}_M$, thus in the rest of the experiments we focus on $\text{BLOOD}_L$. Lastly, while BLOOD demonstrates superior performance on most datasets, the improvements are more consistently observed when applied with ELECTRA compared to RoBERTa. Interestingly, the datasets where BLOOD with RoBERTa outperforms other white/black-box methods (SST, BP, AR, MG, and NG) appear to be more complex, as indicated by the minimum description length (Perez et al., 2021) (cf. Appendix C). We offer explanations for these observations in sections 4.3 and 4.4.

Compared to open-box methods, BLOOD is outperformed by MD in all setups except when using ELECTRA on the TREC dataset. However, BLOOD remains competitive with ENSM and TEMP. Unlike the findings in (Ovadia et al., 2019), the dominance of ENSM is reduced. This is likely because we employ a pre-trained language model ensemble, while they use entirely randomly initialized models. In our ensemble, the model parameters exhibit minimal variation since all models are pre-trained. Variability between models arises solely from the random initialization of the classification head and the stochastic nature of the training process. The high performance of MD on transformer-based language models is aligns with prior research (Podolskiy et al., 2021).

### 4.3 SOURCE OF THE DIFFERENCES IN TRANSFORMATIONS OF ID AND OOD DATA

Understanding which layers of the model are impacted by the model's training could shed some light on the behavior of our method. To find out how much each layer has learned, we examine the changes in intermediate representations of instances after training. For simplicity, we use the Euclidean distances $\|\boldsymbol{r}_{\text{init}} - \boldsymbol{r}_{\text{FT}}\|_2$ between representations of the initialized model ($\boldsymbol{r}_{\text{init}}$) and the representations after fine-tuning the model ($\boldsymbol{r}_{\text{FT}}$). We calculate this distance for all instances in the training set at each of the model's layers and then compute the average for each layer.

Figure 1 illustrates the extent of representation changes in training data alongside BLOOD scores before and after fine-tuning at each intermediate layer. The representations of the upper layers change significantly more than the representations of the lower layers. This is expected since transformer-based language models learn morphological- and syntactic-based features in the lower layers, which are similar between tasks and can be mostly reused from the pre-training. In contrast,

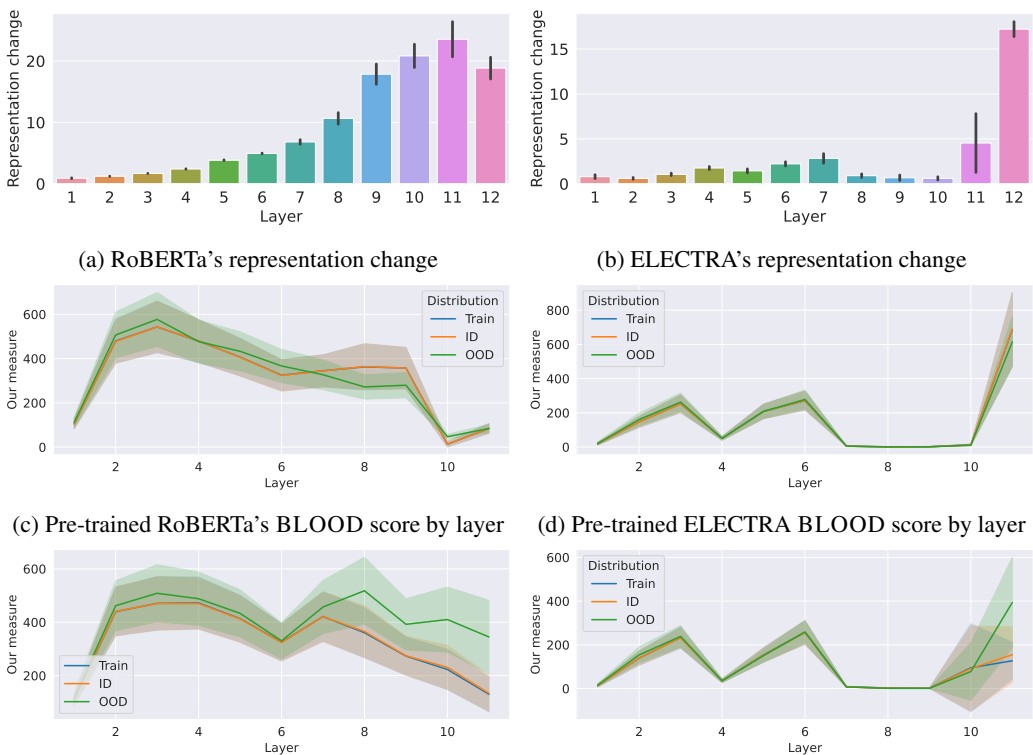

Figure 1: The impact of change of each layer on BLOOD score across layers. Top row: Change in intermediate representations of training instances by layer for (a) RoBERTa and (b) ELECTRA. The scores are averaged across instances for the AR dataset. The black error bars denote the standard deviation. Middle row: BLOOD score by layer of models for AR before fine-tuning. Bottom row: BLOOD score by layer of models for AR after fine-tuning.

higher layers learn more task-specific features such as context and coreferences (Peters et al., 2018; Tenney et al., 2019; Jawahar et al., 2019). Our hypothesis posits that the smooth transformations of ID data are a by-product of the learning algorithm learning the natural progression between abstractions. Consequently, layers more impacted by training will exhibit smoother transformations, which explains why $\mathrm{BLOOD}_L$ outperforms $\mathrm{BLOOD}_M$ on the OOD detection task. This effect becomes apparent when comparing the representation change (upper row of Figure 1) with the BLOOD score (lower two rows of Figure 1) across layers, with a more significant difference in transition smoothness between ID and OOD data observed in layers where representations have undergone more substantial changes overall. The effect is particularly emphasized in ELECTRA, where the last layer undergoes the most significant change, resulting in $\mathrm{BLOOD}_L$ performing exceptionally well due to the radical smoothing of the final transformation.

We also anticipate that the representations of ID data will undergo more significant changes after fine-tuning than those of OOD data, given the model's focus on the ID region of the representation space during training. This effect would cause a difference in smoothness because the ID region of the space would be smoothed out while the OOD region of the space would keep its original sharpness. Same as above, we calculate the change in representations using Euclidean distance of representations before and after fine-tuning. We then quantify the difference between changes in representations of ID and OOD data using the common language effect size (CLES) (McGraw & Wong, 1992), corresponding to the probability that representations of ID data exhibited greater changes after training than representations of OOD data.[4] We measure this difference for the model's last layer and the mean difference across all layers.

---

[4]The CLES statistics quantifies the effect size of the difference between two samples. It is equivalent to AUC of the corresponding univariate binary classifier, representing the probability that a randomly selected score from the first sample will exceed a randomly selected score from the second sample.

Table 3: The performance of OOD detection methods for the simplified datasets measured by AU-ROC (%). The best-performing white/black-box method is in **bold**. Open-box methods that outperform all white/black-box methods are in **bold**. Higher is better. The right side of the table shows a comparison of changes in representations between ID and OOD data using CLES (%).

| Model | Dataset | White-box/Black box | | | | | | | Open-box | | | | CLES | |
|---|---|---|---|---|---|---|---|---|---|---|---|---|---|---|
| | | $\text{BLOOD}_L$ | MSP | ENT | EGY | MC | GRAD | ASH | ReAct | ENSM | TEMP | MD | Mean | Last |
| RoBERTa | BP2 | 79.66 | 89.74 | 89.74 | 88.23 | 88.92 | **89.84** | 82.66 | 87.60 | 87.59 | **89.92** | **97.66** | 94.57 | 84.27 |
| | AR2 | 88.20 | 93.33 | 93.33 | **94.27** | 93.30 | 93.58 | 92.63 | 93.31 | **94.55** | 93.34 | **99.02** | 91.84 | 80.47 |
| | MG2 | 84.78 | 78.13 | 78.13 | **85.44** | 82.62 | 78.28 | 74.05 | **85.95** | 83.95 | 78.23 | **97.48** | 86.80 | 70.25 |
| ELECTRA | BP2 | 71.71 | **93.23** | **93.23** | 92.51 | 92.61 | 93.20 | 86.93 | 92.24 | 91.25 | **93.34** | **98.75** | 97.28 | 94.87 |
| | AR2 | 90.67 | **96.16** | **96.16** | 93.80 | 95.47 | 96.14 | 91.95 | 93.40 | 95.20 | **96.20** | 93.22 | 97.07 | 96.22 |
| | MG2 | **91.41** | 88.02 | 88.02 | 85.08 | 88.55 | 88.10 | 76.48 | 85.11 | 84.12 | 87.95 | **98.28** | 88.28 | 87.10 |

Table 2 shows the effect size quantified using CLES for the changes in representations between ID and OOD data. In most setups, CLES is far above 50%, which means that representations of ID data underwent more significant changes than those of OOD data. The results imply that the learning algorithm's focus during training is on the ID region of the representation space. In contrast, the rest of the representation space is largely unaltered. Moreover, the difference in transformation smoothness between layers, observed between ID and OOD data, can be attributed to the inherently non-smooth transformations of the initialized models. These non-smooth transformations gradually become smoother within the ID region. However, more complex datasets (BP, AR, MG, and NG) in conjunction with the RoBERTa model contradict our initial expectation. In these cases, CLES approaches or even drops below 50%. This indicates that the ID region of the representation space undergoes similar or even lesser changes compared to the rest of the representation space.

Our interpretation of this phenomenon is that the algorithm faces greater difficulty in fitting the data, necessitating more substantial adjustments to the model. These significant alterations not only result in smoothing out transitions for ID data but, as a consequence, also make transformations in the rest of the space less smooth. This would explain the improved performance of BLOOD in conjunction with RoBERTa on these datasets, as the difference in transformation smoothness is attributed not only to the smoothing of the ID region of the space but also to the reduction in smoothness of the remaining space. This sharpening effect in the region populated by OOD data is evident when comparing sub-figures (c) and (e) in Figure 1.

Table 2: Effect size of the changes in representations between ID and OOD data. We calculate CLES (%) averaged across layers (Mean) and for the last layer (Last), showing averages over five random seeds with standard deviation.

| Model | Dataset | Mean | Last |
|---|---|---|---|
| RoBERTa | SST | $66.86 \pm 5.90$ | $63.91 \pm 5.64$ |
| | SUBJ | $78.77 \pm 9.61$ | $68.08 \pm 10.53$ |
| | AGN | $73.28 \pm 3.59$ | $60.18 \pm 4.38$ |
| | TREC | $90.63 \pm 7.19$ | $74.02 \pm 21.03$ |
| | BP | $55.98 \pm 29.52$ | $39.65 \pm 16.38$ |
| | AR | $52.52 \pm 15.53$ | $33.83 \pm 8.45$ |
| | MG | $34.40 \pm 9.56$ | $46.23 \pm 11.90$ |
| | NG | $40.93 \pm 8.51$ | $49.56 \pm 9.14$ |
| ELECTRA | SST | $82.09 \pm 1.31$ | $78.67 \pm 0.97$ |
| | SUBJ | $77.43 \pm 13.52$ | $75.61 \pm 14.63$ |
| | AGN | $81.28 \pm 3.62$ | $80.82 \pm 4.23$ |
| | TREC | $99.86 \pm 0.05$ | $99.10 \pm 0.54$ |
| | BP | $93.35 \pm 2.00$ | $82.80 \pm 3.19$ |
| | AR | $82.21 \pm 9.59$ | $81.95 \pm 7.70$ |
| | MG | $83.88 \pm 6.01$ | $83.83 \pm 7.70$ |
| | NG | $79.08 \pm 8.84$ | $80.16 \pm 4.60$ |

### 4.4 THE EFFECT OF DATASET COMPLEXITY

In the previous subsection, we demonstrated that BLOOD performs better on more complex datasets compared to simpler ones.[5] To investigate this phenomenon further, we re-evaluate the performance of OOD detection methods on simplified versions of the more complex datasets. Specifically, we use the binary classification datasets BP2, AR2, and MG2, which are derived from BP, AR, and MG datasets, respectively, by retaining only two classes (cf. Appendix C for additional details).

Table 3 shows AUROC for the OOD detection task on simplified datasets, as well as the CLES of representation changes. We observe a decrease in AUROC for BLOOD in comparison to the AUROC on the original datasets, while the AUROC of other white/black-box methods shows an increase. The drop in AUROC for BLOOD can be explained by examining the CLES of repre-

---

[5]We support this finding by calculating the Pearson correlation coefficient between MDL and difference in AUROC of $\text{BLOOD}_M$ (to capture the influence on all layers in the model) and the baseline method (MSP) for each dataset. We found a significant ($p < .05$) correlation of 0.79 for RoBERTa and 0.73 for ELECTRA.

Table 4: The performance of OOD detection methods on the task of Near-OOD detection measured by AUROC (%). The best-performing white/black-box method is in **bold**. Open-box methods that outperform all white/black-box methods are in **bold**. Higher is better.

| Model | Shift | White-box/Black-box | | | | | | | Open-box | | | |
|---|---|---|---|---|---|---|---|---|---|---|---|---|
| | | $BLOOD_L$ | MSP | ENT | EGY | MC | GRAD | ASH | ReAct | ENSM | TEMP | MD |
| RoBERTa | Semantic | 61.61 | 69.46 | **69.50** | 69.41 | 68.34 | 69.36 | 66.50 | 69.46 | 68.91 | 70.56 | **72.03** |
| | Background | **62.70** | 54.26 | 54.26 | 50.17 | 48.18 | 54.33 | 50.46 | 49.32 | 49.13 | 54.19 | 59.40 |
| ELECTRA | Semantic | 62.49 | 63.17 | 63.12 | 60.92 | 62.14 | **63.23** | 56.85 | 61.00 | **65.67** | 62.45 | **64.22** |
| | Background | **59.35** | 42.96 | 42.96 | 38.68 | 37.96 | 42.77 | 40.66 | 38.53 | 41.25 | 42.63 | 39.31 |

sentation changes, which exhibits a notable increase compared to the original datasets in the case of RoBERTa, and even a slight increase for ELECTRA. The rise in CLES of the change in representations suggests that the models managed to learn the task without the need to sharpen the transformations of the OOD data, thereby reducing the ability of BLOOD to detect OOD instances.

We suspect that the increase in AUROC for the other white/black-box methods may be attributed to the same factor that led to the AUROC decrease in BLOOD – namely, the task's simplicity. However, this cause manifests differently. The simplified datasets, having fewer ambiguous instances in their test sets due to the reduced number of classes, allow the other (probabilistic) methods to more accurately attribute the estimated uncertainty to the OOD data. See Appendix F for a more detailed explanation and visualization using dataset cartography (Swayamdipta et al., 2020).

### 4.5 TYPES OF DISTRIBUTION SHIFT

Another important aspect to consider for OOD detection is the type of distribution shift. Up to this point, we have only considered OOD data coming from a distribution entirely different than that of the ID data, which is referred to as Far-OOD by Baran et al. (2023). We next examine the performance of OOD detection methods on Near-OOD data, which arises from either a semantic or a background shift. For the semantic shift, in line with Ovadia et al. (2019), we designate the even-numbered classes of NG dataset as ID and the odd-numbered classes as Near-OOD data. For the background shift, following Baran et al. (2023), we use the SST dataset as ID and the Yelp Review sentiment classification dataset (Zhang et al., 2015) as Near-OOD data.

Table 4 shows the OOD detection performance on the semantic and background shift detection tasks. For the semantic shift, BLOOD exhibits suboptimal performance. However, in the case of the background shift, it notably outperforms all other methods, including the open-box approaches, some of which even perform worse than random. We suspect the subpar performance of other OOD detection methods in background shift detection may be attributed to models performing better on Yelp data compared to the SST data they were trained on (cf. Appendix C), because Yelp has longer texts with more semantic cues, making models more confident on OOD data. We speculate the discrepancy in performance between semantic and background shifts arises because BLOOD is focused on the encoding process of the query instances, while other methods only examine the model's outputs. Consequently, BLOOD demonstrates greater sensitivity to the changes in the data-generating distribution. At the same time, other methods are better at detecting changes in the outputs, such as the introduction of an unknown class. In Appendix G we show that BLOOD is sensitive to the degree of distribution shift.

## 5 CONCLUSION

We have proposed a novel method for out-of-distribution (OOD) detection for Transformer-based networks called BLOOD. The method analyzes representation transformations across intermediate layers and requires only the access to model's weights. Our evaluation on multiple text classification datasets using Transformer-based large pre-trained language models shows that BLOOD outperforms similar methods. Our analysis reveals that ID representations undergo smoother transformations between layers compared to OOD representations because the model concentrates on the ID region of the representation space during training. We demonstrated that the learning algorithm retains the original sharpness of the transformations of OOD intermediate representations for simpler datasets but increases the sharpness for more complex datasets.

ACKNOWLEDGMENT

We thank the anonymous reviewers for their insightful comments. Our heartfelt appreciation goes to the members of TakeLab for their continuous support and valuable input. Special thanks to Nina Drobac and Stjepan Šebek for their feedback and helpful suggestions. This work has been supported by the Croatian Science Foundation under the project IP-2020-02-8671 PSYTXT ("Computational Models for Text-Based Personality Prediction and Analysis").

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

## A    REPRODUCIBILITY

We conducted our experiments on 4× AMD Ryzen Threadripper 3970X 32-Core Processors and 2x NVIDIA GeForce RTX 3090 GPUs with 24GB of RAM, which took a little bit less than three weeks. We used Python 3.8.5, PyTorch (Paszke et al., 2019) version 1.12.1, Hugging Face Transformers (Wolf et al., 2020) version 4.21.3, Hugging Face Datasets (Lhoest et al., 2021) version 2.11.0, scikit-learn (Pedregosa et al., 2011) version 1.2.2, and CUDA 11.4.

## B    PROOF OF THE COROLLARY

**Corollary 1.** *Let $J(x) \in \mathbb{R}^{m \times n}$ be a Jacobian matrix, and let $\mathbf{v} \in \mathbb{R}^n$ and $\mathbf{w} \in \mathbb{R}^m$ be random vectors whose elements are independent random variables with zero mean and unit variance. Then, $\mathbb{E}[(\mathbf{w}^\intercal J(x)\mathbf{v})^2] = \|J(x)\|_F^2$.*

**Remark 1.** *Clearly, the result holds true regardless of whether $J(x)$ is a Jacobian matrix of some transformation or, indeed, any constant matrix $A$ with real entries. Henceforth, we assume the latter is the case.*

It turns out that the requirements on random vectors $\mathbf{v}$ and $\mathbf{w}$ can be relaxed, and a more general statement from which Corollary 1 trivially follows is given in Theorem 1 below:

**Theorem 1.** *Let $A \in \mathbb{R}^{m \times n}$ be a constant $m \times n$ matrix, and let $\mathbf{v} \in \mathbb{R}^n$ and $\mathbf{w} \in \mathbb{R}^m$ be independent random vectors with identity autocorrelation matrices $\mathbb{E}[\mathbf{vv}^\intercal] = I_n$ and $\mathbb{E}[\mathbf{ww}^\intercal] = I_m$, then $\mathbb{E}[(\mathbf{w}^\intercal A\mathbf{v})^2] = \|A\|_F^2$.*

Before the proof of this theorem, we will need to show two lemmas. The first one says that if one wishes to find a sum of squared 2-norms of $n$ vectors $\sum_i \|\mathbf{a}_i\|_2^2$ (which one may interpret as the square of the Frobenius norm $\|A\|_F^2$ of the matrix obtained by putting all those vectors together), one can do this stochastically by taking a random linear combination of those vectors and then squaring the 2-norm of the resulting vector $\|\sum_i \mathrm{v}_i\mathbf{a}_i\|_2^2$. The random weights have to satisfy $\mathbb{E}[\mathrm{v}_i\mathrm{v}_j] = \delta_{ij}$, where $\delta_{ij}$ is Kronecker delta which is 1 if $i = j$ and 0 otherwise.

**Lemma 1.** *Let $A \in \mathbb{R}^{m \times n}$ be a constant $m \times n$ matrix, and let $\mathbf{v} \in \mathbb{R}^n$ be a random vector with identity autocorrelation matrix $\mathbb{E}[\mathbf{vv}^\intercal] = I_n$. Then, $\mathbb{E}[\|A\mathbf{v}\|_2^2] = \|A\|_F^2$.*

*Proof.* Denote by $\mathbf{a}_i$ columns of matrix $A$, so that $A = [\mathbf{a}_1|\dots|\mathbf{a}_n]$. The matrix-vector product

$$A\mathbf{v} = \sum_i \mathrm{v}_i\mathbf{a}_i$$

where $\mathrm{v}_i$ denote entries of the vector $\mathbf{v}$. Note

$$\|A\mathbf{v}\|_2^2 = (A\mathbf{v})^\intercal A\mathbf{v} = \left(\sum_i \mathrm{v}_i\mathbf{a}_i\right)^\intercal \left(\sum_j \mathrm{v}_j\mathbf{a}_j\right) = \left(\sum_i \mathrm{v}_i\mathbf{a}_i^\intercal\right) \left(\sum_j \mathrm{v}_j\mathbf{a}_j\right)$$

$$= \sum_{i,j} \mathbf{a}_i^\intercal\mathbf{a}_j\mathrm{v}_i\mathrm{v}_j.$$

Therefore,

$$\mathbb{E}[\|A\mathbf{v}\|_2^2] = \mathbb{E}[\sum_{i,j} \mathbf{a}_i^\intercal\mathbf{a}_j\mathrm{v}_i\mathrm{v}_j] = \sum_{i,j} \mathbf{a}_i^\intercal\mathbf{a}_j\mathbb{E}[\mathrm{v}_i\mathrm{v}_j] = \sum_i \mathbf{a}_i^\intercal\mathbf{a}_i = \sum_i \|\mathbf{a}_i\|_2^2 = \|A\|_F^2.$$

$\square$

The next lemma deals with the same principle as the previous lemma, but this time for scalars rather than vectors. In short, if one wishes to find a sum of squares of $m$ scalars $\sum_j \mathrm{a}_j^2$ (which one may interpret as the square of the 2-norm of the vector $\mathbf{a}$ with those components), one can do this stochastically by taking a random linear combination of those scalars and then just squaring the sum $(\sum_j \mathrm{w}_j\mathrm{a}_j)^2$. Again, the random weights have to satisfy $\mathbb{E}[\mathrm{w}_i\mathrm{w}_j] = \delta_{ij}$.

**Lemma 2.** *Let $\mathbf{a} \in \mathbb{R}^m$ be a constant row-vector, and let $\mathbf{w} \in \mathbb{R}^m$ be a random vector with identity autocorrelation matrix $\mathbb{E}[\mathbf{w}\mathbf{w}^\intercal] = \boldsymbol{I}_m$. Then, $\mathbb{E}[(\mathbf{a}^\intercal\mathbf{w})^2] = \|\mathbf{a}\|_2^2$.*

*Proof.* This is a direct consequence of Lemma 1. Just take $\boldsymbol{A} = \mathbf{a}^\intercal$ to be a row-vector and note that the Frobenius norm of that row-vector is just its Euclidean 2-norm. □

*Proof of Theorem 1.* When conditioning on $\mathbf{v}$, $\boldsymbol{A}\mathbf{v}$ is a constant vector and we can use Lemma 2 to write:
$$\|\boldsymbol{A}\mathbf{v}\|_2^2 = \mathbb{E}[((\boldsymbol{A}\mathbf{v})^\intercal\mathbf{w})^2 \mid \mathbf{v}] = \mathbb{E}[(\mathbf{v}^\intercal\boldsymbol{A}^\intercal\mathbf{w})^2 \mid \mathbf{v}] = \mathbb{E}[(\mathbf{w}^\intercal\boldsymbol{A}\mathbf{v})^2 \mid \mathbf{v}].$$
where in the last step we transposed the $1 \times 1$ matrix $\mathbf{v}^\intercal\boldsymbol{A}^\intercal\mathbf{w}$. Putting this together with Lemma 1 gives:
$$\|\boldsymbol{A}\|_F^2 = \mathbb{E}[\|\boldsymbol{A}\mathbf{v}\|_2^2] = \mathbb{E}[\mathbb{E}[(\mathbf{w}^\intercal\boldsymbol{A}\mathbf{v})^2 \mid \mathbf{v}]] = \mathbb{E}[(\mathbf{w}^\intercal\boldsymbol{A}\mathbf{v})^2].$$
□

**Remark 2.** *The estimates above are closely related to the so-called Hutchinson's trick (Hutchinson, 1990), which gives an unbiased estimate of the trace of a matrix as*
$$\mathrm{tr}(\boldsymbol{W}) = \mathbb{E}[\mathbf{v}^\intercal\boldsymbol{W}\mathbf{v}],$$
*where $\mathbf{v}$ satisfies the same conditions as before. Our estimate in Lemma 1 can be seen as its corollary since*
$$\|\boldsymbol{A}\|_F^2 = \mathrm{tr}(\boldsymbol{A}^\intercal\boldsymbol{A}) = \mathbb{E}[\mathbf{v}^\intercal\boldsymbol{A}^\intercal\boldsymbol{A}\mathbf{v}] = \mathbb{E}[\|\boldsymbol{A}\mathbf{v}\|_2^2].$$
*The proof of Theorem 1 is not new. It has appeared in (Bujanovic & Kressner, 2021), although the result was stated there in lesser generality. We decided to include both the proofs of Lemma 1 and Theorem 1 for completeness.*

**Remark 3.** *Note that both Lemma 1 and Theorem 1 estimate $\|\boldsymbol{A}\|_F^2$. It is possible to show that the variance of the estimator in Lemma 1 is bounded above by the variance of the estimator in Theorem 1. This should be intuitively clear as the latter takes the vector $\boldsymbol{A}\mathbf{v}$ and rather than just taking its exact 2-norm (like the former), it further projects it onto another random vector $\mathbf{w}$ in order to estimate its 2-norm (cf. Proof of Theorem 1)*

**Remark 4.** *The estimate given in Theorem 1 is most useful when both dimensions of the matrix $\boldsymbol{M}$ are large, and if obtaining its entries is computationally expensive, but calculating the vector-matrix-vector product can be performed efficiently. If, in addition, one can perform matrix-vector product efficiently (which is the case when, e.g., $m$ is small) it is beneficial to use the estimator given in Lemma 1. The same is true (by transposing everything) if $n$ is small and/or one can perform vector-matrix product efficiently.*

## C EXPERIMENTAL DESIGN

In this section, we present details about the models and datasets used in the experiments along with the description and hyperparamters of the training procedures to ensure the reproducibility of the results.

### C.1 MODELS

For our experiments, we choose two popular transformer-based large pre-trained language models that obtain state-of-the-art results on a variety of NLP tasks, e.g., text classification, named entity recognition, machine translation, text summarization. Both models have similar architecture with the main difference being the pre-training data and pre-training objectives. We used the same hyperparamters for both models. For fine-tuning we used Adam optimizer (Kingma & Ba, 2015) with $\beta_1 = 0.9$, $\beta_2 = 0.999$, $\epsilon = 10^{-8}$, learning rate of $2 \times 10^{-5}$. We fine-tuned the models for ten epochs. The batch size depends on the dataset used. We repeated each experiment with five different random seeds that varied the initialization of the classification head and the stochastic nature of the learning procedure. The models we use are:

- **RoBERTa** – Uses masked language modeling (MLM) pre-training objective. The model has 12 layers, a hidden state size of 768, and 12 attention heads with 125M parameters in total;

- **ELECTRA** – Unlike RobERTa which uses generative pre-training objective, ELECTRA uses discriminative pre-training objective. The model has 12 layers, a hidden state size of 768, and 12 attention heads with 110M parameters in total.

## C.2 DATASETS

In our experiments, we work with several text classification datasets. Datasets we used as ID data and their preprocessing procedures are:

- SST – The Stanford Sentiment Treebank dataset contains single sentences extracted from movie reviews labeled with the sentiment of the review. The task is an almost balanced binary classification with labels corresponding to positive and negative sentiment;

- SUBJ – The Subjectivity dataset is a collection of movie review documents. The task is to classify the reviews into one of two balanced classes based on the nature of the review: objective or subjective;

- AGN – The AG News topic classification dataset consists of news articles from several news sources. The dataset consists of four balanced classes representing the topic of the article: World, Sports, Business, and Sci/Tech. The train split of the dataset was subsampled to 20000 instances for our experiments keeping the balance of the labels;

- TREC – The Text REtrieval Conference (TREC) Question Classification dataset gathers questions labeled with their topics: Abbreviation, Entity, Description and abstract concept, Human being, Location, and Numeric value. The dataset contains six imbalanced labels;

- BP – BigPatent consists of records of U.S. patent descriptions along with human written abstractive summaries from nine patent categories: Human Necessities, Performing Operations and Transporting, Chemistry and Metallurgy, Textiles and Paper, Fixed Constructions, Mechanical Engineering and Lightning and Heating and Weapons and Blasting, Physics, Electricity, and General tagging of new or cross-sectional technology. Even though this dataset is usually used for summarization we use the summaries for classification. We remove all duplicates from train and test splits and between train and test splits. We subsample the whole dataset by taking 3% of the original train set and 20% of the original test set;

- AR – Amazon Customer Reviews dataset contains customer reviews of products from Amazon store. The dataset contains the product category along with the review text. We subset all of the categories to 12 with comparable sizes and significant semantic differences: Gift Card, Software, Video Games, Luggage, Video, Grocery, Furniture, Musical Instruments, Watches, Tools, Baby, and Jewelry. We preprocess the data first dropping all of the reviews with less than 15 words and all of the duplicates from the training split. We then subsample 0.25% of the data and split that subsample into train and test sets with sizes of 80% and 20% of the subsample size, respectively;

- MG – IMDb Genre Classification Dataset is used for the classification of movies' genres from their descriptions from IMDb. In our experiment, we use a subset of 15 biggest genres with significant semantic differences: Drama, Documentary, Comedy, Horror, Thriller, Action, Western, Reality TV, Adventure, Family, Music, Romance, Sci-fi, Adult, and Crime. We preprocess the data by first removing all duplicates from train and test splits and between train and test splits, and then subsample the train data to 50% of the original size and test data to 15% of the original size;

- NG – 20Newsgroups data is a collection of news documents labeled based on the topic of the news with 20 different labels that are almost uniformly distributed. Following sci-kit-learn, we preprocess both train and test data by removing headers, signature blocks, and quotation blocks to eliminate simple correlations to which models easily overfit. We also remove any potential duplicate documents between train and test sets to avoid data leaks.

Simplified datasets were preprocessed the same as their original counterparts described above. After the preprocesing, datasets were simplified by removing the data from all of the labels but two from both train and test sets. The choice of the retained two labels was made on the basis of the absolute and relative sizes of the data with given labels and semantic differences between labels.

Table 5: Information about the datasets used in the experiments. The table contains sizes of the train and ID test sets, number of classes in each dataset, micro $F_1$ score on the test set for RoBERTa and ELECTRA, minimum description length (MDL) as an estimate of data complexity (Based on Rissanen Data Analysis. We take an average of five runs with RoBERTa and ELECTRA. MDL is computed with 128 blocks of 32 instances per batch, Adam optimizer and a learning rate of $2 \times 10^{-5}$. Numbers in the table are normalized with respect to the largest value.), and batch size used to train the model.

| Dataset | Train | Test | # Labels | $F_1^{\text{RoBERTa}}$ | $F_1^{\text{ELECTRA}}$ | MDL | Batch Size |
|---|---|---|---|---|---|---|---|
| SST | 8544 | 2210 | 2 | $87.70 \pm 1.17$ | $88.60 \pm 0.50$ | 0.389 | 32 |
| SUBJ | 8000 | 2000 | 2 | $96.30 \pm 0.49$ | $96.83 \pm 0.62$ | 0.148 | 32 |
| AGN | 20000 | 7600 | 4 | $92.63 \pm 0.35$ | $92.18 \pm 0.80$ | 0.279 | 16 |
| TREC | 5452 | 500 | 6 | $96.92 \pm 0.37$ | $96.56 \pm 0.73$ | 0.138 | 32 |
| BP | 20792 | 7678 | 9 | $64.24 \pm 0.67$ | $63.87 \pm 0.92$ | 1 | 16 |
| AR | 20336 | 5085 | 12 | $87.26 \pm 0.40$ | $86.65 \pm 0.70$ | 0.324 | 16 |
| MG | 22832 | 6849 | 15 | $70.99 \pm 0.57$ | $71.32 \pm 0.44$ | 0.652 | 16 |
| NG | 11314 | 7306 | 20 | $72.65 \pm 0.37$ | $71.38 \pm 0.44$ | 0.751 | 16 |
| BP2 | 7410 | 2735 | 2 | $93.52 \pm 0.22$ | $93.61 \pm 0.50$ | 0.253 | 16 |
| AR2 | 6388 | 1597 | 2 | $97.27 \pm 0.59$ | $97.85 \pm 0.27$ | 0.124 | 16 |
| MG2 | 13291 | 3986 | 2 | $93.38 \pm 0.36$ | $93.30 \pm 0.28$ | 0.281 | 16 |

- **BP2** – BigPatent2 retains labels Human Necessities and Physics.
- **AR2** – AmazonReviews2 retains labels Grocery and Baby.
- **MG2** – MovieGenre2 retains labels Drama and Documentary.

Datasets we used as OOD data and their preprocessing procedures are:

- **OBW** – One Billion Word Benchmark is a popular dataset for training and evaluating language models. In this paper, we use it as the OOD data for that reason, i.e., the diversity of the corpus. We use the test set of the corpus and in each experiment, we subsample it to the size of the ID test set.
- **Yelp** – The Yelp Reviews dataset consists of reviews from Yelp labeled for the sentiment of the review, e.g., positive or negative. The task is balanced binary sentiment classification. When using this dataset as OOD data we subsample the test set of the dataset to the size of the ID test set. Transfer of RoBERTa model fine-tuned on SST data achieves $F_1$ score of $94.13 \pm 0.79$, while transfer of ELECTRA model fine-tuned on SST data achieves $F_1$ score of $95.05 \pm 1.24$.

More details about the used datasets are shown in Table 5.

## D    BLOOD FOR IMAGE CLASSIFICATION

To explore the effectiveness of our BLOOD methodology on image data, we utilized CIFAR-10 and CIFAR-100 as our ID datasets (Krizhevsky et al., 2009). These were chosen for their resemblance to the ImageNet dataset, which was used to pre-train the Vision Transformer (ViT) (Dosovitskiy et al., 2021). We tested OOD detection using two different datasets: Street View House Numbers (SVHN; Yuval, 2011), which comprises real-world imagery of house numbers offering different visual features from CIFAR datasets, and the Beans dataset, which contains images relevant to agricultural disease classification, adding another layer of diversity to our evaluation.

In addition to the Transformer architecture, we also employ convolutional neural networks (CNNs), specifically ResNet34 and ResNet50 (He et al., 2016), to gauge how our method performs on other architectures. For CNNs, we employed a learning rate of $0.01$ and trained the models over 50 epochs. In contrast, for our transformer-based model, the ViT, we adopted a fine-tuning approach, utilizing a learning rate of $2 \times 10^{-5}$ with an additional weight decay of $0.001$, over a shorter span of 10 epochs. This distinction in training methodologies is reflective of the different learning dynamics and capacities of CNNs versus Transformers.

Table 6: The performance of OOD detection methods on the task of OOD detection on image classification measured by AUROC (%). The best-performing method is in **bold**. Higher is better.

| Model | ID Dataset | $BLOOD_M$ | $BLOOD_L$ | MSP | ENT | EGY | MC | GRAD | ASH |
|---|---|---|---|---|---|---|---|---|---|
| | | | | | SVHN | | | | |
| ResNet34 | CIFAR-10 | 84.00 | 49.99 | 88.92 | **89.71** | 88.23 | 81.39 | 82.29 | 83.15 |
| | CIFAR-100 | **87.05** | 49.51 | 73.64 | 76.20 | 77.90 | 78.31 | 85.97 | 80.54 |
| ResNet50 | CIFAR-10 | 84.13 | 49.91 | 89.55 | **90.29** | 84.36 | 90.07 | 88.01 | 88.20 |
| | CIFAR-100 | 79.77 | 50.50 | 79.86 | 83.32 | 82.70 | **83.42** | 81.80 | 80.89 |
| ViT | CIFAR-10 | 99.37 | 92.45 | 99.12 | 98.19 | **99.55** | 98.91 | 96.94 | 95.01 |
| | CIFAR-100 | 80.97 | **89.07** | 84.06 | 84.58 | 88.21 | 87.47 | 85.53 | 81.29 |
| | | | | | Beans | | | | |
| ResNet34 | CIFAR-10 | 85.68 | 14.16 | 89.23 | 91.55 | **92.41** | 91.09 | 81.04 | 90.67 |
| | CIFAR-100 | **86.40** | 23.54 | 73.32 | 79.90 | 81.22 | 83.97 | 76.55 | 81.02 |
| ResNet50 | CIFAR-10 | 78.37 | 50.28 | 79.18 | **80.45** | 79.02 | 79.62 | 69.33 | 77.26 |
| | CIFAR-100 | **86.62** | 49.76 | 79.13 | 80.79 | 82.31 | 83.07 | 85.29 | 75.16 |
| ViT | CIFAR-10 | 95.41 | 99.58 | 99.31 | 99.31 | **99.98** | 98.79 | 96.92 | 94.31 |
| | CIFAR-100 | 91.68 | **96.53** | 96.02 | 95.71 | 95.81 | 95.83 | 89.33 | 88.14 |

Table 7: Performance of BLOOD in an open-box setting. Cases in which the open-box BLOOD ouperforms both $BLOOD_M$ and $BLOOD_L$ are in **bold**.

| Model | SST | SUBJ | AGN | TREC | BP | AR | MG | NG |
|---|---|---|---|---|---|---|---|---|
| RoBERTa | **73.94** | **82.98** | **81.39** | 91.73 | **93.47** | **96.25** | **92.46** | **89.97** |
| ELECTRA | 77.67 | 77.30 | **82.76** | 98.73 | 96.54 | **91.97** | **90.92** | **85.19** |

We show the results in Table 6. The ViT model, with its transformer architecture, showed a marked improvement in AUROC with $BLOOD_L$, underscoring the critical role of the last layer in such models. This aligns with our observations from textual data analysis, where the most significant changes also occurred in the last layer, suggesting a pattern that transformers exhibit across different data modalities.

Conversely, the CNN models, ResNet34 and ResNet50, displayed more nuanced changes across their layers. Since these models were trained from scratch, the learning occurred more prominently in the lower layers, and the last layer often had an inverse impact on OOD detection capabilities. This was evidenced by low AUROC scores for $BLOOD_L$, while $BLOOD_M$ remained competitive with other methods. Additionally, BLOOD's ability to leverage the complexity of datasets was apparent both with ViT and CNNs, particularly with the more granular CIFAR-100 dataset compared to CIFAR-10, which is consistent with the observations on textual data.

# E  BLOOD AS AN OPEN-BOX METHOD

In the scenarios where more resources are available, beyond just the trained model, it is possible to utilize those resources to improve the OOD detection performance of BLOOD. Similarly to a lot of popular OOD detection methods (Liang et al., 2018; Sun et al., 2021; Sun & Li, 2022), BLOOD can be improved by using a small validation set to learn the optimal weights for the weighted average of the BLOOD score in each layer.

To obtain the weights for the weighted sum, first we create a validation set from the 5% of our ID and OOD test sets and then fit the logistic regression. In Table 7 we showcase the results of using BLOOD in the open-box scenarios. From the results it can be seen that it is useful to extend BLOOD to open-box setting if the validation set is obtainable. In rare cases $BLOOD_L$ outperform the open-box BLOOD likely due to noise introduced by including the lower layers. But still, the differences in performance when open-box BLOOD is outperformed by the $BLOOD_L$ are minuscule compared to the gains in other cases.

## F    Dataset Complexity and Uncertainty

The identification of OOD instances often relies on estimating the underlying uncertainty of the model (Ovadia et al., 2019). The intuition is that a well-performing model should exhibit higher confidence when dealing with data resembling the training data, i.e., ID data. However, ML models are susceptible to two sources of uncertainty. *Aleatoric* uncertainty stems from the inherent ambiguity and noise in the data, and it is thus irreducible by the acquisition of more data and characteristic of ambiguous ID data; In contrast, *epistemic* uncertainty stems from the lack of relevant information in the training data, and it is thus reducible by acquiring more relevant data and characteristic of the OOD data (Der Kiureghian & Ditlevsen, 2009; Kendall & Gal, 2017). The total amount of uncertainty about a given prediction, i.e., aleatoric and epistemic uncertainty, is called *predictive uncertainty* (Gal, 2016).

In our experiments, we use probabilistic baselines such as MSP, ENT, and MC. However, these approaches cannot reliably disentangle the epistemic uncertainty from the model's predictive uncertainty (Kirsch et al., 2021).

In Section 4.4 we show our method, BLOOD, performs better on complex datasets than on simpler datasets. The results of our experiment also show the opposite of that for probabilistic methods (MSP, ENT, and MC), i.e., they work better on simpler datasets than on more complex datasets. We hypothesize that the effect of probabilistic methods working better on simpler datasets comes from the amount of aleatoric uncertainty in the dataset. Because of their simplicity, simpler datasets have less ambiguous instances, and thus with lowered aleatoric uncertainty, epistemic uncertainty of OOD instances dominates the model's predictive uncertainty.

To visualize the drop in ambiguity, we use data cartography (Swayamdipta et al., 2020). The idea is to use the training dynamics of the examples to discover easy-to-learn, hard-to-learn, and ambiguous examples in the dataset. Training dynamics used to create data maps are confidence (how confident the model is in the true label across epochs), variability (the spread of the posterior probability of the true label across epochs), and correctness (the fraction of times the model correctly labels the example across epochs). Ambiguous examples are characterized by high variability, and it can be seen from Figure 2 that simplified datasets, by removing classes, lowered the density of ambiguous examples in the ID dataset. Since the other white/black-box methods can not disentangle aleatoric from epistemic uncertainty, lowering the density of ambiguous examples helps them capture the epistemic uncertainty in the OOD data needed for their detection.

## G    Degree of Distribution Shift

Another important feature of an OOD detection method is the proportional sensitivity to the degree of distribution shift involved in the data (Ovadia et al., 2019). In Figure 3, we show that the uncertainty scores produced by $BLOOD_L$ increase in proportion to the degree of distribution shift. Training data exhibits the lowest uncertainty score, ID test data shows only a slight increase in uncertainty, Near-OOD data exhibits a jump in the uncertainty score, while for Far-OOD data, the uncertainty score is the highest.

## H    Additional Results

In Section 4 we present the results of our experiments measured with AUROC averaged across five different seeds. In this section, we present averaged results alongside their standard deviations for AUROC as well as two other commonly used metrics in the OOD detection literature, AUPR-IN and FPR@95TPR:

- **AUPR-IN** – area under the Precision-Recall curve illustrates how precision and recall vary with different thresholds of OOD detection method's uncertainty score. The ID data is considered a positive class. A higher score indicates better performance.

- **FPR@95TPR** – the false positive rate of an OOD classifier when the true positive rate is 95%, The ID data is considered a positive class. A lower score indicates better performance.

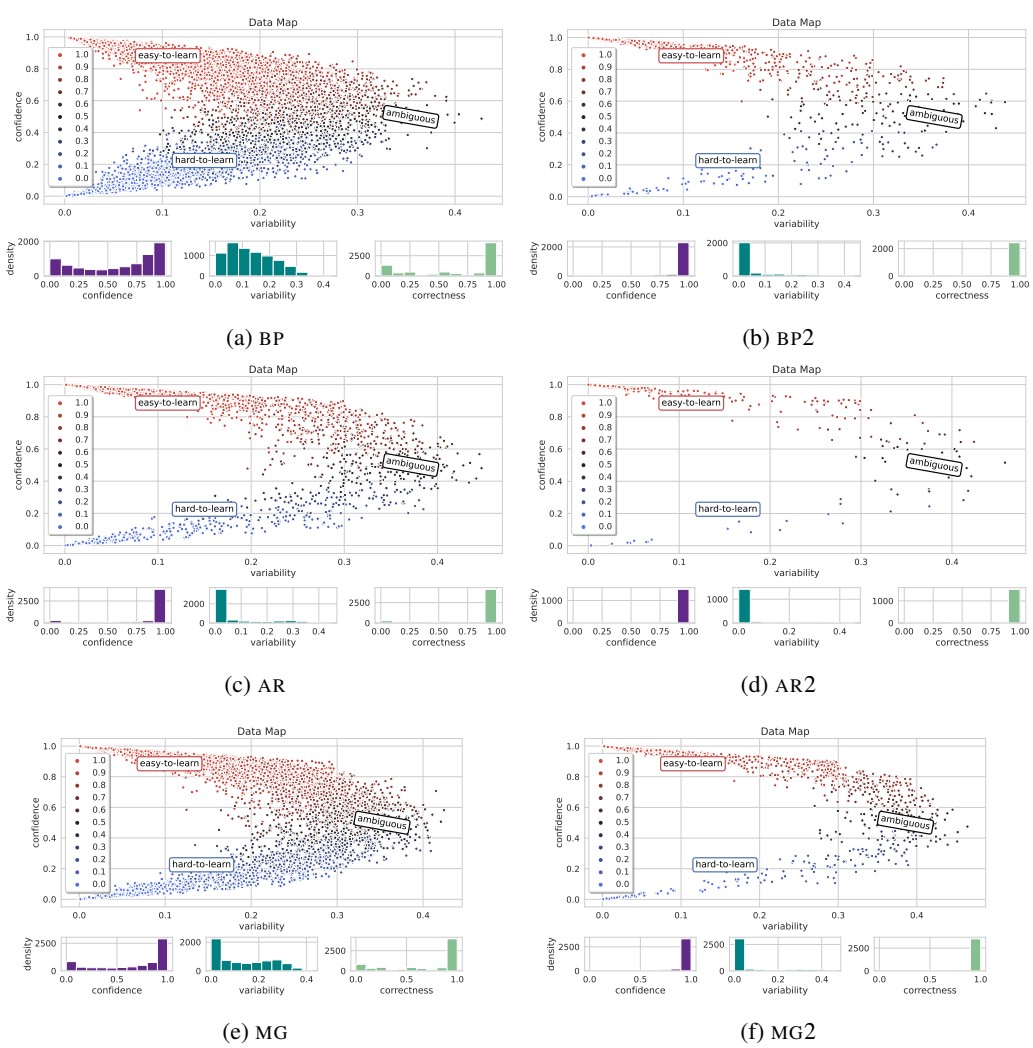

Figure 2: Data maps with RoBERTa for test sets of (a) BP, (b) BP2, (c) AR, AR2, MG, and MG2. Each subfigure shows data map and histograms of confidence, variability, and correctness of instances. Data maps for ELECTRA are qualitatively the same.

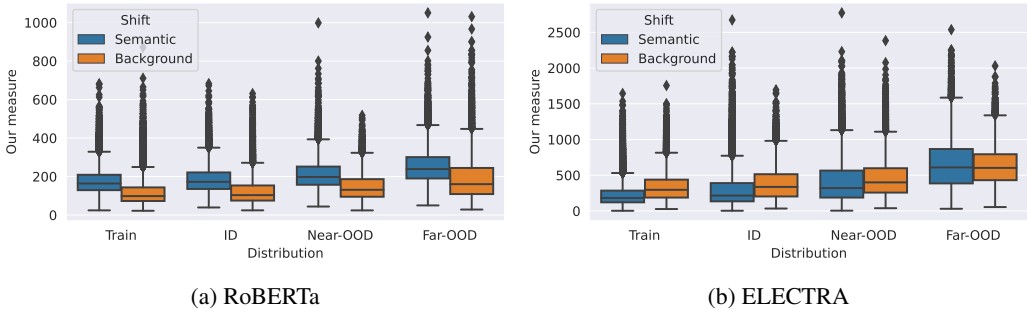

Figure 3: Box plots of change in $BLOOD_L$ scores with an increase in the degree of distribution shift for the tasks of semantic and background shift detection for (a) RoBERTa and (b) ELECTRA. The amount of distribution shift increases from left to right: training distribution, test ID data distribution, Near-OOD distribution, and Far-OOD distribution.

Table 8: OOD detection performance measured by AUROC (%) with standard deviation of the white-box/black-box methods. The best-performing measure is in **bold**. Higher is better.

| Model | Dataset | BLOOD$_M$ | BLOOD$_L$ | MSP | ENT | EGY | MC | GRAD |
|---|---|---|---|---|---|---|---|---|
| RoBERTa | SST | 50.56 ± 7.03 | **72.83 ± 9.72** | 71.61 ± 0.84 | 71.69 ± 1.38 | 71.69 ± 1.38 | 68.28 ± 1.16 | 71.76 ± 1.34 |
| | SUBJ | 52.02 ± 14.01 | 74.66 ± 7.32 | **75.79 ± 8.95** | 74.55 ± 8.48 | 74.55 ± 8.48 | 74.21 ± 6.82 | 74.93 ± 8.57 |
| | AGN | 77.46 ± 5.73 | 61.95 ± 8.87 | 76.36 ± 3.34 | 73.57 ± 2.96 | 73.8 ± 2.99 | **77.55 ± 3.04** | 73.58 ± 2.94 |
| | TREC | 69.63 ± 10.11 | 95.3 ± 2.96 | 96.28 ± 0.74 | 96.2 ± 0.83 | **96.40 ± 0.79** | 95.68 ± 0.78 | 96.14 ± 0.84 |
| | BP | 87.20 ± 2.95 | **89.53 ± 3.37** | 85.84 ± 1.34 | 70.15 ± 0.84 | 72.82 ± 0.98 | 74.29 ± 1.06 | 73.11 ± 1.49 |
| | AR | 91.41 ± 1.78 | **93.20 ± 1.24** | 92.39 ± 0.77 | 89.06 ± 1.11 | 89.96 ± 1.05 | 90.59 ± 0.92 | 88.65 ± 1.06 |
| | MG | **88.15 ± 1.87** | 85.23 ± 3.99 | 86.45 ± 2.87 | 75.02 ± 2.33 | 76.6 ± 2.47 | 79.98 ± 2.34 | 74.28 ± 2.27 |
| | NG | **83.53 ± 1.13** | 72.02 ± 4.76 | 82.65 ± 0.73 | 77.49 ± 0.73 | 78.76 ± 0.82 | 79.32 ± 1.10 | 76.93 ± 0.75 |
| ELECTRA | SST | 74.36 ± 2.77 | **78.11 ± 2.09** | 71.97 ± 1.72 | 73.84 ± 1.89 | 73.84 ± 1.89 | 70.81 ± 2.01 | 73.82 ± 1.92 |
| | SUBJ | 74.1 ± 11.02 | 77.41 ± 11.41 | 70.46 ± 7.16 | **78.17 ± 7.78** | **78.17 ± 7.78** | 77.71 ± 8.40 | 78.11 ± 7.86 |
| | AGN | 65.67 ± 6.90 | **80.28 ± 3.41** | 79.75 ± 3.96 | 76.8 ± 3.18 | 77.01 ± 3.28 | 79.55 ± 2.74 | 76.57 ± 3.15 |
| | TREC | 97.48 ± 0.84 | **98.90 ± 0.37** | 97.48 ± 0.65 | 97.26 ± 0.93 | 97.56 ± 0.82 | 96.21 ± 0.79 | 97.07 ± 0.96 |
| | BP | 86.06 ± 1.85 | **96.72 ± 1.4** | 84.63 ± 1.80 | 78.56 ± 2.57 | 81.75 ± 2.58 | 83.04 ± 2.33 | 76.77 ± 3.04 |
| | AR | 84.58 ± 3.49 | **91.66 ± 2.59** | 90.64 ± 1.64 | 87.74 ± 1.68 | 88.44 ± 1.76 | 88.53 ± 2.16 | 87.52 ± 1.62 |
| | MG | 80.52 ± 7.30 | **90.63 ± 3.56** | 80.41 ± 4.15 | 73.83 ± 4.12 | 74.78 ± 4.25 | 76.67 ± 3.46 | 73.35 ± 4.04 |
| | NG | 77.61 ± 2.12 | **82.47 ± 2.85** | 80.83 ± 2.88 | 76.45 ± 2.66 | 77.73 ± 2.74 | 79.11 ± 2.16 | 75.97 ± 2.63 |

Table 9: OOD detection performance measured by AUROC (%) with standard deviation of the open-box measures. Measures that outperform all of the white-box/black-box methods are in **bold**. Higher is better.

| Model | Dataset | ENSM | TEMP | MD |
|---|---|---|---|---|
| RoBERTa | SST | 69.03 ± 1.07 | 71.64 ± 1.60 | **85.36 ± 3.42** |
| | SUBJ | **76.68 ± 1.91** | 74.41 ± 8.60 | **93.47 ± 0.66** |
| | AGN | **80.35 ± 1.15** | 75.38 ± 2.93 | **82.63 ± 3.19** |
| | TREC | **96.87 ± 0.54** | **96.74 ± 0.84** | **96.74 ± 2.01** |
| | BP | 79.39 ± 1.07 | 86.01 ± 1.02 | **97.35 ± 1.12** |
| | AR | 92.44 ± 0.26 | 92.25 ± 0.85 | **98.35 ± 0.26** |
| | MG | 76.98 ± 1.54 | 84.3 ± 3.2 | **95.12 ± 1.68** |
| | NG | 80.77 ± 1.2 | 82.87 ± 0.49 | **90.68 ± 0.95** |
| ELECTRA | SST | 73.81 ± 1.18 | 73.58 ± 1.98 | **78.85 ± 1.48** |
| | SUBJ | **79.23 ± 3.17** | **78.20 ± 7.87** | **81.59 ± 8.16** |
| | AGN | 79.5 ± 2.03 | 78.31 ± 3.59 | **86.1 ± 1.85** |
| | TREC | 97.55 ± 0.37 | 98.2 ± 0.57 | 97.54 ± 1.18 |
| | BP | 84.2 ± 1.68 | 84.69 ± 2.12 | **98.28 ± 0.46** |
| | AR | **91.98 ± 1.08** | 90.35 ± 1.7 | **95.47 ± 0.83** |
| | MG | 76.86 ± 1.08 | 78.47 ± 4.66 | **92.96 ± 3.67** |
| | NG | 79.93 ± 0.83 | 80.75 ± 2.69 | **89.13 ± 0.86** |

Results of OOD detection measured with AUROC are given in Table 8 for white-box/black-box methods and in Table 9 for open-box methods. Results for simplified datasets measured with AUROC are given in Table 10 for white-box/black-box methods and in Table 11 for open-box methods. Results of distribution Nera-OOD detection measured with AUROC are given in Table 12 for white-box/black-box methods and in Table 13 for open-box methods.

Results of OOD detection measured with AUPR are given in Table 14 for white-box/black-box methods and in Table 15 for open-box methods. Results for simplified datasets measured with AUPR are given in Table 16 for white-box/black-box methods and in Table 17 for open-box methods. Results of distribution Near-OOD detection measured with AUPR are given in Table 18 for white-box/black-box methods and in Table 19 for open-box methods.

Results of OOD detection measured with FPR@95TPR are given in Table 20 for white-box/black-box methods and in Table 21 for open-box methods. Results for simplified datasets measured with FPR@95TPR are given in Table 22 for white-box/black-box methods and in Table 23 for open-box methods. Results of distribution Near-OOD detection measured with FPR@95TPR are given in Table 24 for white-box/black-box methods and in Table 25 for open-box methods.

We also provide visualization of the assessment of how changes in individual layers influence the BLOOD score throughout intermediate layers. Figures 4, 5, 6, 7, 8, 9, and 10 show vizualizations for SST, SUBJ, AGN, TREC BP, MG, and NG respectively akin to Figure 1 for AR.

Table 10: OOD detection performance measured by AUROC (%) with standard deviation of the simplified datasets. The best-performing measure is in **bold**. Higher is better.

| Model | Dataset | BLOOD$_L$ | MSP | ENT | EGY | MC | GRAD |
|---|---|---|---|---|---|---|---|
| RoBERTa | BP2 | $79.66 \pm 10.03$ | $89.74 \pm 1.20$ | $89.74 \pm 1.20$ | $88.23 \pm 0.72$ | $88.92 \pm 1.33$ | $\mathbf{89.84 \pm 1.13}$ |
| | AR2 | $88.20 \pm 5.04$ | $93.33 \pm 3.98$ | $93.33 \pm 3.98$ | $\mathbf{94.27 \pm 1.00}$ | $93.30 \pm 2.72$ | $93.58 \pm 3.75$ |
| | MG2 | $84.78 \pm 14.37$ | $78.13 \pm 6.36$ | $78.13 \pm 6.36$ | $\mathbf{85.44 \pm 3.25}$ | $82.62 \pm 4.25$ | $78.28 \pm 6.37$ |
| ELECTRA | BP2 | $71.71 \pm 16.41$ | $\mathbf{93.23 \pm 2.00}$ | $\mathbf{93.23 \pm 2.00}$ | $92.51 \pm 1.45$ | $92.61 \pm 1.44$ | $93.20 \pm 2.02$ |
| | AR2 | $90.67 \pm 5.55$ | $\mathbf{96.16 \pm 0.89}$ | $\mathbf{96.16 \pm 0.89}$ | $93.80 \pm 3.80$ | $95.47 \pm 1.01$ | $96.14 \pm 0.90$ |
| | MG2 | $\mathbf{91.41 \pm 1.45}$ | $88.02 \pm 2.15$ | $88.02 \pm 2.15$ | $85.08 \pm 6.62$ | $88.55 \pm 1.52$ | $88.10 \pm 2.11$ |

Table 11: OOD detection performance measured by AUROC (%) with standard deviation of the open-box measures for the simplified dataset is shown left of the vertical line. Measures that outperform all white-box/black-box methods are in **bold**. Higher is better. Effect size of changes in representations between ID and OOD data for a simplified datasets using CLES (%) is shown right of the vertical line.

| Model | Dataset | ENSM | TEMP | MD | Mean | Last |
|---|---|---|---|---|---|---|
| RoBERTa | BP2 | $87.59 \pm 1.30$ | $\mathbf{89.92 \pm 1.18}$ | $\mathbf{97.66 \pm 0.61}$ | $94.57 \pm 6.33$ | $84.27 \pm 3.85$ |
| | AR2 | $\mathbf{94.55 \pm 0.69}$ | $93.34 \pm 4.03$ | $\mathbf{99.02 \pm 0.29}$ | $91.84 \pm 4.70$ | $80.47 \pm 6.56$ |
| | MG2 | $83.95 \pm 1.79$ | $78.23 \pm 6.29$ | $\mathbf{97.48 \pm 0.83}$ | $86.8 \pm 11.25$ | $70.25 \pm 10.69$ |
| ELECTRA | | $91.25 \pm 1.10$ | $\mathbf{93.34 \pm 2.01}$ | $\mathbf{98.75 \pm 0.33}$ | $94.87 \pm 1.21$ | |
| | AR2 | $95.20 \pm 1.04$ | $\mathbf{96.20 \pm 0.91}$ | $93.22 \pm 6.26$ | $97.07 \pm 1.01$ | $96.22 \pm 2.63$ |
| | MG2 | $84.12 \pm 2.71$ | $87.95 \pm 2.09$ | $\mathbf{98.28 \pm 0.34}$ | $88.28 \pm 1.09$ | $87.1 \pm 2.65$ |

Table 12: Near-OOD detection performance measured by AUROC (%) with standard deviation of the white-box/black-box methods. The best-performing measure is in **bold**. Higher is better.

| Model | Shift | BLOOD$_L$ | MSP | ENT | EGY | MC | GRAD |
|---|---|---|---|---|---|---|---|
| RoBERTa | semantic | $61.61 \pm 2.61$ | $69.46 \pm 0.83$ | $69.41 \pm 0.99$ | $\mathbf{69.50 \pm 0.86}$ | $68.34 \pm 0.56$ | $69.36 \pm 0.91$ |
| | background | $\mathbf{62.7 \pm 5.75}$ | $54.26 \pm 2.49$ | $50.17 \pm 4.82$ | $54.26 \pm 2.49$ | $48.18 \pm 2.44$ | $54.33 \pm 2.5$ |
| ELECTRA | semantic | $62.49 \pm 3.81$ | $63.17 \pm 2.65$ | $60.92 \pm 3.70$ | $63.12 \pm 2.69$ | $62.14 \pm 2.26$ | $\mathbf{63.23 \pm 2.63}$ |
| | background | $\mathbf{59.35 \pm 3.19}$ | $42.96 \pm 2.92$ | $38.68 \pm 2.25$ | $42.96 \pm 2.92$ | $37.96 \pm 3.47$ | $42.77 \pm 2.92$ |

Table 13: Near-OOD detection performance measured by AUROC (%) with standard deviation of the open-box measures for the augmented dataset is shown left of the vertical line. Measures that outperform all white-box/black-box methods are in **bold**. Higher is better.

| Model | Shift | ENSM | TEMP | MD |
|---|---|---|---|---|
| RoBERTa | semantic | $68.91 \pm 1.12$ | $\mathbf{70.56 \pm 1.25}$ | $\mathbf{72.03 \pm 0.89}$ |
| | background | $49.13 \pm 2.5$ | $54.19 \pm 2.57$ | $59.4 \pm 10.18$ |
| ELECTRA | semantic | $\mathbf{65.67 \pm 0.45}$ | $62.45 \pm 3.17$ | $\mathbf{64.22 \pm 2.75}$ |
| | background | $41.25 \pm 2.47$ | $42.63 \pm 2.84$ | $39.31 \pm 4.93$ |

Table 14: OOD detection performance measured by AUPR-IN (%) with standard deviation of the white-box/black-box methods. The best-performing measure is in **bold**. Higher is better.

| Model | Dataset | BLOOD$_M$ | BLOOD$_L$ | MSP | ENT | EGY | MC | GRAD |
|---|---|---|---|---|---|---|---|---|
| RoBERTa | SST | $50.72 \pm 5.49$ | $\mathbf{72.68 \pm 9.50}$ | $71.13 \pm 1.35$ | $71.49 \pm 2.49$ | $71.49 \pm 2.49$ | $70.35 \pm 2.04$ | $71.59 \pm 2.44$ |
| | SUBJ | $54.60 \pm 11.38$ | $73.45 \pm 8.69$ | $73.68 \pm 11.08$ | $76.18 \pm 7.48$ | $76.18 \pm 7.47$ | $75.85 \pm 6.76$ | $\mathbf{76.66 \pm 7.54}$ |
| | AGN | $\mathbf{76.48 \pm 5.49}$ | $59.48 \pm 8.20$ | $72.73 \pm 3.70$ | $71.42 \pm 3.78$ | $71.47 \pm 3.78$ | $74.64 \pm 3.93$ | $71.26 \pm 3.64$ |
| | TREC | $67.70 \pm 9.83$ | $94.98 \pm 3.18$ | $96.31 \pm 0.76$ | $96.67 \pm 0.88$ | $\mathbf{96.78 \pm 0.85}$ | $96.29 \pm 0.83$ | $96.64 \pm 0.87$ |
| | BP | $86.80 \pm 3.01$ | $\mathbf{89.98 \pm 2.83}$ | $84.50 \pm 0.76$ | $69.72 \pm 0.88$ | $71.06 \pm 0.98$ | $72.58 \pm 0.85$ | $73.29 \pm 1.15$ |
| | AR | $91.30 \pm 1.92$ | $\mathbf{93.18 \pm 1.23}$ | $92.32 \pm 0.84$ | $89.95 \pm 1.17$ | $90.35 \pm 1.19$ | $91.25 \pm 1.08$ | $89.75 \pm 1.16$ |
| | MG | $\mathbf{87.86 \pm 1.94}$ | $84.18 \pm 4.69$ | $86.31 \pm 2.93$ | $77.69 \pm 2.57$ | $78.36 \pm 2.65$ | $81.04 \pm 2.26$ | $77.27 \pm 2.55$ |
| | NG | $\mathbf{84.70 \pm 0.99}$ | $72.97 \pm 4.44$ | $82.59 \pm 0.99$ | $78.74 \pm 0.80$ | $79.29 \pm 0.84$ | $80.11 \pm 0.92$ | $78.43 \pm 0.81$ |
| ELECTRA | SST | $73.50 \pm 3.25$ | $\mathbf{79.28 \pm 1.66}$ | $69.72 \pm 1.92$ | $73.75 \pm 1.97$ | $73.75 \pm 1.97$ | $72.15 \pm 2.11$ | $73.69 \pm 2.03$ |
| | SUBJ | $73.62 \pm 11.47$ | $78.02 \pm 11.63$ | $70.99 \pm 7.04$ | $79.86 \pm 6.71$ | $79.86 \pm 6.71$ | $\mathbf{79.93 \pm 7.30}$ | $79.78 \pm 6.84$ |
| | AGN | $64.00 \pm 5.74$ | $\mathbf{78.77 \pm 3.04}$ | $77.13 \pm 4.37$ | $75.63 \pm 3.46$ | $75.71 \pm 3.48$ | $78.00 \pm 2.98$ | $75.37 \pm 3.46$ |
| | TREC | $97.39 \pm 0.84$ | $\mathbf{98.89 \pm 0.40}$ | $97.34 \pm 0.62$ | $97.78 \pm 0.75$ | $97.94 \pm 0.70$ | $96.73 \pm 0.74$ | $97.66 \pm 0.78$ |
| | BP | $87.87 \pm 1.89$ | $\mathbf{96.54 \pm 1.41}$ | $82.67 \pm 3.05$ | $79.34 \pm 3.26$ | $80.97 \pm 3.31$ | $82.81 \pm 2.66$ | $77.16 \pm 3.53$ |
| | AR | $86.27 \pm 2.79$ | $\mathbf{91.95 \pm 3.04}$ | $90.81 \pm 1.38$ | $88.69 \pm 1.38$ | $89.05 \pm 1.43$ | $89.73 \pm 1.57$ | $88.53 \pm 1.31$ |
| | MG | $81.65 \pm 6.40$ | $\mathbf{91.14 \pm 3.30}$ | $79.82 \pm 4.52$ | $75.49 \pm 4.64$ | $75.91 \pm 4.68$ | $77.99 \pm 3.63$ | $75.21 \pm 4.58$ |
| | NG | $78.90 \pm 3.08$ | $\mathbf{82.37 \pm 5.11}$ | $79.20 \pm 4.28$ | $76.85 \pm 3.50$ | $77.42 \pm 3.63$ | $79.08 \pm 3.10$ | $76.60 \pm 3.53$ |

Table 15: OOD detection performance measured by AUPR-IN (%) with standard deviation of the open-box measures. Measures that outperform all of the white-box/black-box methods are in **bold**. Higher is better.

| Model | Dataset | ENSM | TEMP | MD |
|---|---|---|---|---|
| RoBERTa | SST | 69.91 ± 1.79 | 66.91 ± 3.13 | **85.77 ± 3.49** |
| | SUBJ | **78.51 ± 3.20** | 72.32 ± 8.40 | **93.53 ± 0.75** |
| | AGN | **77.45 ± 1.62** | 67.88 ± 3.91 | **79.93 ± 3.90** |
| | TREC | **97.28 ± 0.46** | 96.39 ± 1.12 | **96.91 ± 1.85** |
| | BP | 77.44 ± 1.01 | 80.61 ± 1.35 | **96.93 ± 1.28** |
| | AR | 92.73 ± 0.34 | 90.37 ± 1.22 | **98.24 ± 0.27** |
| | MG | 77.62 ± 1.81 | 81.20 ± 3.85 | **95.14 ± 1.51** |
| | NG | 80.83 ± 1.24 | 79.15 ± 0.94 | **91.54 ± 0.84** |
| ELECTRA | SST | 75.15 ± 1.40 | 69.21 ± 2.20 | **81.24 ± 0.94** |
| | SUBJ | **81.71 ± 3.41** | 76.71 ± 7.49 | **82.03 ± 6.97** |
| | AGN | 77.76 ± 3.20 | 72.91 ± 3.95 | **83.90 ± 3.48** |
| | TREC | 97.93 ± 0.24 | 98.02 ± 0.66 | 97.60 ± 1.12 |
| | BP | 82.91 ± 2.00 | 79.52 ± 3.31 | **98.27 ± 0.43** |
| | AR | **92.42 ± 0.98** | 88.61 ± 1.66 | **96.09 ± 0.72** |
| | MG | 78.34 ± 1.32 | 74.60 ± 5.69 | **93.67 ± 3.27** |
| | NG | 80.01 ± 1.20 | 75.91 ± 4.15 | **90.97 ± 0.80** |

Table 16: OOD detection performance measured by AUPR-IN (%) with standard deviation of the simplified datasets. The best-performing measure is in **bold**. Higher is better.

| Model | Dataset | BLOOD$_L$ | MSP | ENT | EGY | MC | GRAD |
|---|---|---|---|---|---|---|---|
| RoBERTa | BP2 | 78.96 ± 10.76 | 90.37 ± 0.95 | 90.37 ± 0.95 | 89.03 ± 1.35 | **90.55 ± 0.96** | 90.50 ± 0.88 |
| | AR2 | 86.43 ± 6.29 | 92.81 ± 5.97 | 92.81 ± 5.97 | **94.68 ± 0.71** | 93.82 ± 3.79 | 93.24 ± 5.32 |
| | MG2 | 82.94 ± 15.31 | 77.55 ± 7.63 | 77.55 ± 7.63 | **85.45 ± 3.94** | 83.11 ± 4.93 | 77.84 ± 7.45 |
| ELECTRA | BP2 | 72.95 ± 15.63 | 93.22 ± 3.21 | 93.22 ± 3.21 | 93.05 ± 1.83 | **93.42 ± 2.26** | 93.15 ± 3.27 |
| | AR2 | 88.97 ± 6.73 | **96.34 ± 1.20** | **96.34 ± 1.20** | 94.32 ± 3.57 | 96.20 ± 1.06 | 96.30 ± 1.25 |
| | MG2 | 91.12 ± 1.00 | 90.34 ± 1.33 | 90.34 ± 1.33 | 86.75 ± 6.33 | **91.22 ± 0.80** | 90.40 ± 1.30 |

Table 17: OOD detection performance measured by AUPR-IN (%) with standard deviation of the open-box measures for the simplified dataset is shown left of the vertical line. Measures that outperform all white-box/black-box methods are in **bold**. Higher is better.

| Model | Dataset | ENSM | TEMP | MD |
|---|---|---|---|---|
| RoBERTa | BP2 | 89.20 ± 1.39 | 88.70 ± 1.20 | **97.94 ± 0.58** |
| | AR2 | **95.79 ± 0.68** | 91.42 ± 7.25 | **99.10 ± 0.23** |
| | MG2 | **84.69 ± 3.85** | 74.03 ± 8.73 | **97.73 ± 0.78** |
| ELECTRA | BP2 | 93.03 ± 2.09 | 92.01 ± 3.83 | **98.81 ± 0.29** |
| | AR2 | **96.45 ± 0.79** | 95.64 ± 1.41 | 87.32 ± 12.53 |
| | MG2 | 87.43 ± 2.78 | 88.64 ± 1.47 | **98.49 ± 0.30** |

Table 18: Near-OOD detection performance measured by AUPR-IN (%) with standard deviation of the white-box/black-box methods. The best-performing measure is in **bold**. Higher is better.

| Model | Shift | BLOOD$_L$ | MSP | ENT | EGY | MC | GRAD |
|---|---|---|---|---|---|---|---|
| RoBERTa | semantic | 61.15 ± 3.20 | **70.56 ± 2.81** | 70.36 ± 2.87 | 70.00 ± 1.29 | 69.99 ± 2.58 | 70.42 ± 2.88 |
| | background | 62.62 ± 5.66 | 56.61 ± 4.48 | 56.61 ± 4.48 | 51.12 ± 5.68 | 52.43 ± 4.43 | 56.68 ± 4.46 |
| ELECTRA | semantic | 61.28 ± 5.53 | 61.85 ± 2.25 | 61.76 ± 2.32 | 58.57 ± 3.85 | 60.62 ± 1.85 | **61.97 ± 2.20** |
| | background | 59.40 ± 3.31 | 45.36 ± 2.09 | 45.36 ± 2.09 | 42.00 ± 1.11 | 41.66 ± 2.21 | 45.26 ± 2.09 |

Table 19: Near-OOD detection performance measured by AUPR-IN (%) with standard deviation of the open-box measures for the augmented dataset is shown left of the vertical line. Measures that outperform all white-box/black-box methods are in **bold**. Higher is better.

| Model | Shift | ENSM | TEMP | MD |
|---|---|---|---|---|
| RoBERTa | semantic | 69.50 ± 2.84 | 67.13 ± 3.45 | **73.76 ± 1.27** |
| | background | 52.33 ± 3.19 | 51.30 ± 5.11 | **65.79 ± 8.60** |
| ELECTRA | semantic | **63.49 ± 1.69** | 55.54 ± 3.31 | **64.94 ± 4.82** |
| | background | 43.50 ± 1.38 | 39.75 ± 2.00 | 46.59 ± 4.32 |

Table 20: OOD detection performance measured by FPR@95TPR (%) with standard deviation of the white-box/black-box methods. The best-performing measure is in **bold**. Lower is better.

| Model | Dataset | BLOOD$_M$ | BLOOD$_L$ | MSP | ENT | EGY | MC | GRAD |
|---|---|---|---|---|---|---|---|---|
| RoBERTa | SST | 94.11 ± 3.57 | **79.31 ± 10.96** | 86.92 ± 0.51 | 86.92 ± 0.51 | 86.03 ± 2.29 | 90.90 ± 1.01 | 87.35 ± 0.76 |
| | SUBJ | 92.65 ± 5.85 | 75.88 ± 10.04 | 80.65 ± 6.14 | 80.65 ± 6.14 | **75.74 ± 5.77** | 85.40 ± 3.52 | 80.81 ± 6.08 |
| | AGN | 70.76 ± 10.35 | 83.21 ± 8.49 | 77.45 ± 3.08 | 76.16 ± 3.09 | **63.96 ± 5.94** | 68.23 ± 3.90 | 77.52 ± 2.94 |
| | TREC | 77.16 ± 13.86 | 21.96 ± 11.91 | 19.72 ± 5.79 | 18.84 ± 5.82 | **18.52 ± 3.56** | 28.20 ± 5.40 | 20.08 ± 5.38 |
| | BP | 49.75 ± 8.85 | **46.77 ± 14.68** | 80.81 ± 1.77 | 71.10 ± 2.79 | 51.05 ± 5.75 | 68.82 ± 2.80 | 83.60 ± 1.85 |
| | AR | 38.04 ± 7.21 | **31.80 ± 5.90** | 59.68 ± 3.95 | 49.18 ± 3.13 | 36.88 ± 3.59 | 47.31 ± 3.19 | 62.43 ± 3.52 |
| | MG | **47.51 ± 5.60** | 52.45 ± 8.99 | 81.20 ± 2.20 | 71.80 ± 3.80 | 50.76 ± 7.53 | 65.02 ± 4.57 | 84.63 ± 1.69 |
| | NG | **67.91 ± 3.33** | 83.79 ± 5.22 | 78.91 ± 1.31 | 72.84 ± 0.99 | 70.04 ± 2.40 | 73.91 ± 1.99 | 81.92 ± 1.42 |
| ELECTRA | SST | 79.75 ± 3.94 | **77.66 ± 4.07** | 84.10 ± 0.79 | 84.10 ± 0.79 | 83.97 ± 2.57 | 89.30 ± 0.98 | 84.09 ± 0.93 |
| | SUBJ | 77.48 ± 10.00 | **75.27 ± 11.62** | 79.67 ± 9.49 | 79.67 ± 9.49 | 82.89 ± 4.86 | 82.80 ± 6.09 | 79.70 ± 9.40 |
| | AGN | 85.07 ± 11.95 | **63.68 ± 15.00** | 77.97 ± 4.64 | 76.93 ± 5.28 | 62.94 ± 9.51 | 71.32 ± 5.52 | 78.48 ± 4.46 |
| | TREC | 11.20 ± 4.06 | **3.84 ± 1.65** | 12.28 ± 5.22 | 11.40 ± 4.81 | 12.28 ± 3.75 | 24.64 ± 7.74 | 13.24 ± 5.70 |
| | BP | 73.77 ± 12.84 | **14.86 ± 6.85** | 76.73 ± 3.31 | 62.47 ± 4.02 | 53.56 ± 3.40 | 59.49 ± 6.27 | 79.19 ± 4.12 |
| | AR | 73.06 ± 14.25 | **42.24 ± 7.35** | 64.30 ± 4.51 | 57.35 ± 5.90 | 45.68 ± 7.24 | 56.56 ± 7.00 | 85.25 ± 2.35 |
| | MG | 75.77 ± 13.56 | **43.06 ± 13.99** | 82.90 ± 2.70 | 75.95 ± 4.32 | 62.25 ± 6.70 | 71.83 ± 4.26 | 65.19 ± 4.16 |
| | NG | 86.26 ± 4.66 | 76.89 ± 4.50 | 78.84 ± 3.79 | 73.33 ± 4.38 | 70.89 ± 4.64 | **72.52 ± 3.5** | 81.76 ± 3.16 |

Table 21: OOD detection performance measured by FPR@95TPR (%) with standard deviation of the open-box measures. Measures that outperform all of the white-box/black-box methods are in **bold**. Lower is better.

| Model | Dataset | ENSM | TEMP | MD |
|---|---|---|---|---|
| RoBERTa | SST | 90.97 ± 0.45 | 86.95 ± 0.37 | **59.63 ± 10.21** |
| | SUBJ | 87.23 ± 1.52 | 80.49 ± 5.95 | **32.60 ± 3.68** |
| | AGN | **62.09 ± 1.84** | 67.71 ± 4.12 | **55.12 ± 6.12** |
| | TREC | 20.40 ± 5.49 | **18.00 ± 4.53** | **16.04 ± 10.83** |
| | BP | 62.59 ± 2.78 | **46.33 ± 3.58** | **11.33 ± 4.96** |
| | AR | 37.70 ± 1.82 | 36.06 ± 3.52 | **7.74 ± 1.67** |
| | MG | 68.95 ± 1.85 | 53.77 ± 7.40 | **23.20 ± 7.71** |
| | NG | 70.41 ± 1.44 | **67.54 ± 1.43** | **49.95 ± 5.09** |
| ELECTRA | SST | 88.62 ± 1.95 | 84.21 ± 0.59 | 84.45 ± 4.86 |
| | SUBJ | 83.87 ± 2.13 | 79.17 ± 10.15 | **68.15 ± 13.19** |
| | AGN | 68.27 ± 2.85 | 70.61 ± 8.14 | **46.47 ± 4.98** |
| | TREC | 18.48 ± 4.99 | 8.60 ± 2.74 | 9.12 ± 5.98 |
| | BP | 53.33 ± 3.32 | 52.09 ± 4.81 | **8.31 ± 2.46** |
| | AR | **41.31 ± 5.67** | 46.08 ± 6.96 | **27.52 ± 8.00** |
| | MG | 71.18 ± 1.23 | 66.34 ± 6.76 | **34.45 ± 14.27** |
| | NG | **72.27 ± 1.35** | **69.58 ± 5.00** | **62.98 ± 4.75** |

Table 22: OOD detection performance measured by FPR@95TPR (%) with standard deviation of the simplified datasets. The best-performing measure is in **bold**. Lower is better.

| Model | Dataset | BLOOD$_L$ | MSP | ENT | EGY | MC | GRAD |
|---|---|---|---|---|---|---|---|
| RoBERTa | BP2 | 62.28 ± 17.17 | 55.45 ± 8.77 | 55.45 ± 8.77 | 64.56 ± 5.95 | 66.65 ± 5.39 | **55.01 ± 8.48** |
| | AR2 | 39.75 ± 15.50 | 29.69 ± 10.78 | 29.69 ± 10.78 | 30.78 ± 8.40 | 39.32 ± 12.32 | **29.12 ± 10.95** |
| | MG2 | **41.38 ± 23.22** | 73.73 ± 8.16 | 73.73 ± 8.16 | 67.98 ± 6.38 | 72.90 ± 5.66 | 73.30 ± 8.07 |
| ELECTRA | BP2 | 65.40 ± 27.16 | 34.58 ± 9.20 | 34.58 ± 9.20 | 44.50 ± 7.54 | 49.86 ± 9.84 | **34.54 ± 9.16** |
| | AR2 | 30.29 ± 15.97 | **15.99 ± 4.57** | **15.99 ± 4.57** | 32.97 ± 26.91 | 26.73 ± 10.82 | 16.02 ± 4.55 |
| | MG2 | **36.59 ± 7.84** | 68.92 ± 8.81 | 68.92 ± 8.81 | 66.01 ± 9.33 | 69.22 ± 7.05 | 68.40 ± 8.61 |

Table 23: OOD detection performance measured by FPR@95TPR (%) with standard deviation of the open-box measures for the simplified dataset is shown left of the vertical line. Measures that outperform all white-box/black-box methods are in **bold**. Lower is better.

| Model | Dataset | ENSM | TEMP | MD |
|---|---|---|---|---|
| RoBERTa | BP2 | 70.41 ± 3.73 | 54.22 ± 8.01 | **11.53 ± 3.33** |
| | AR2 | 41.87 ± 5.09 | **28.90 ± 11.07** | **4.68 ± 1.62** |
| | MG2 | 75.55 ± 1.77 | 73.57 ± 8.19 | **12.83 ± 5.26** |
| ELECTRA | BP2 | 63.95 ± 4.01 | **33.65 ± 8.64** | **6.60 ± 2.34** |
| | AR2 | 34.50 ± 14.15 | **15.84 ± 5.08** | 13.32 ± 7.92 |
| | MG2 | 76.87 ± 4.14 | 68.90 ± 8.82 | **8.53 ± 2.86** |

Table 24: Near-OOD detection performance measured by FPR@95TPR (%) with standard deviation of the white-box/black-box methods. The best-performing measure is in **bold**. Lower is better.

| Model | Shift | BLOOD$_L$ | MSP | ENT | EGY | MC | GRAD |
|---|---|---|---|---|---|---|---|
| RoBERTa | semantic | $89.79 \pm 1.75$ | $90.21 \pm 0.76$ | $89.89 \pm 0.88$ | $\mathbf{89.59 \pm 0.90}$ | $91.50 \pm 0.83$ | $90.21 \pm 0.93$ |
| | background | $\mathbf{91.38 \pm 5.83}$ | $95.32 \pm 1.07$ | $95.32 \pm 1.07$ | $95.79 \pm 1.38$ | $96.79 \pm 0.72$ | $95.25 \pm 1.03$ |
| ELECTRA | semantic | $91.14 \pm 0.92$ | $90.99 \pm 0.74$ | $\mathbf{90.98 \pm 0.98}$ | $91.72 \pm 0.94$ | $91.64 \pm 0.90$ | $\mathbf{90.98 \pm 0.47}$ |
| | background | $\mathbf{89.38 \pm 2.40}$ | $96.30 \pm 0.88$ | $96.30 \pm 0.88$ | $97.12 \pm 0.32$ | $97.64 \pm 0.73$ | $96.31 \pm 0.78$ |

Table 25: Near-OOD detection performance measured by FPR@95TPR (%) with standard deviation of the open-box measures for the augmented dataset is shown left of the vertical line. Measures that outperform all white-box/black-box methods are in **bold**. Lower is better.

| Model | Shift | ENSM | TEMP | MD |
|---|---|---|---|---|
| RoBERTa | semantic | $90.39 \pm 0.49$ | $\mathbf{88.77 \pm 1.03}$ | $\mathbf{84.49 \pm 1.83}$ |
| | background | $97.05 \pm 0.57$ | $95.34 \pm 1.06$ | $95.57 \pm 3.69$ |
| ELECTRA | semantic | $\mathbf{90.26 \pm 0.15}$ | $91.23 \pm 1.26$ | $93.37 \pm 0.29$ |
| | background | $97.71 \pm 0.33$ | $96.27 \pm 0.95$ | $98.83 \pm 0.30$ |

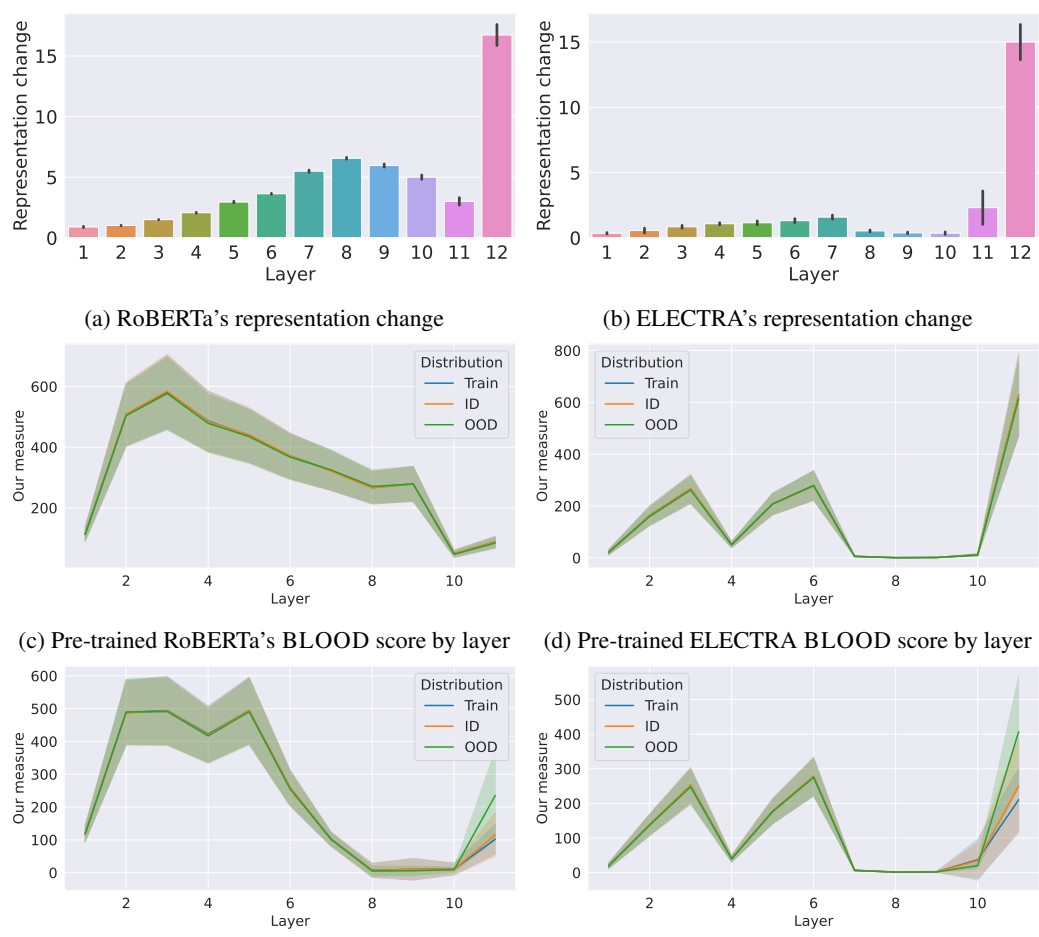

(a) RoBERTa's representation change      (b) ELECTRA's representation change

(c) Pre-trained RoBERTa's BLOOD score by layer      (d) Pre-trained ELECTRA BLOOD score by layer

(e) Fine-tuned RoBERTa's BLOOD score by layer      (f) Fine-tuned ELECTRA BLOOD score by layer

Figure 4: The impact of change of each layer on BLOOD score across layers. Top row: Change in intermediate representations of training instances by layer for (a) RoBERTa and (b) ELECTRA. The scores are averaged across instances for the SST dataset. The black error bars denote the standard deviation. Middle row: BLOOD score by layer of models for SST before fine-tuning. Bottom row: BLOOD score by layer of models for SST after fine-tuning.

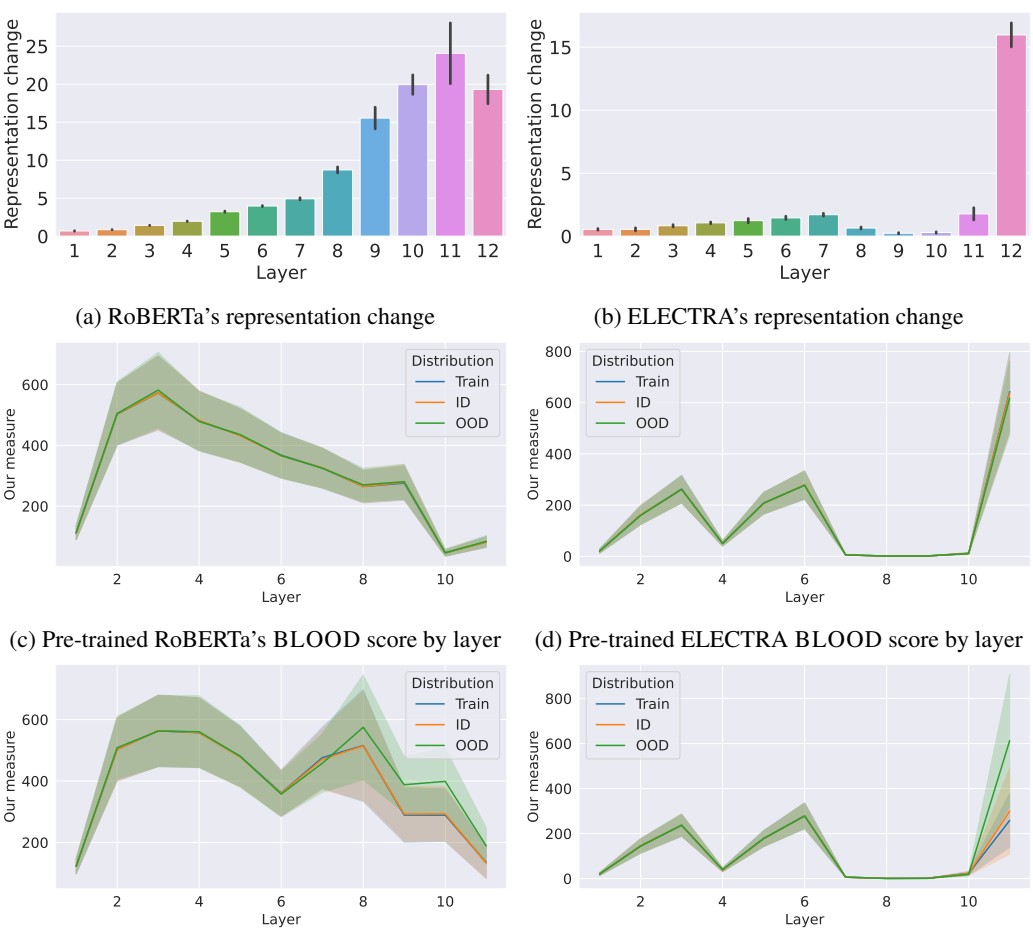

(a) RoBERTa's representation change

(b) ELECTRA's representation change

(c) Pre-trained RoBERTa's BLOOD score by layer

(d) Pre-trained ELECTRA BLOOD score by layer

(e) Fine-tuned RoBERTa's BLOOD score by layer

(f) Fine-tuned ELECTRA BLOOD score by layer

Figure 5: The impact of change of each layer on BLOOD score across layers. Top row: Change in intermediate representations of training instances by layer for (a) RoBERTa and (b) ELECTRA. The scores are averaged across instances for the SUBJ dataset. The black error bars denote the standard deviation. Middle row: BLOOD score by layer of models for SUBJ before fine-tuning. Bottom row: BLOOD score by layer of models for SUBJ after fine-tuning.

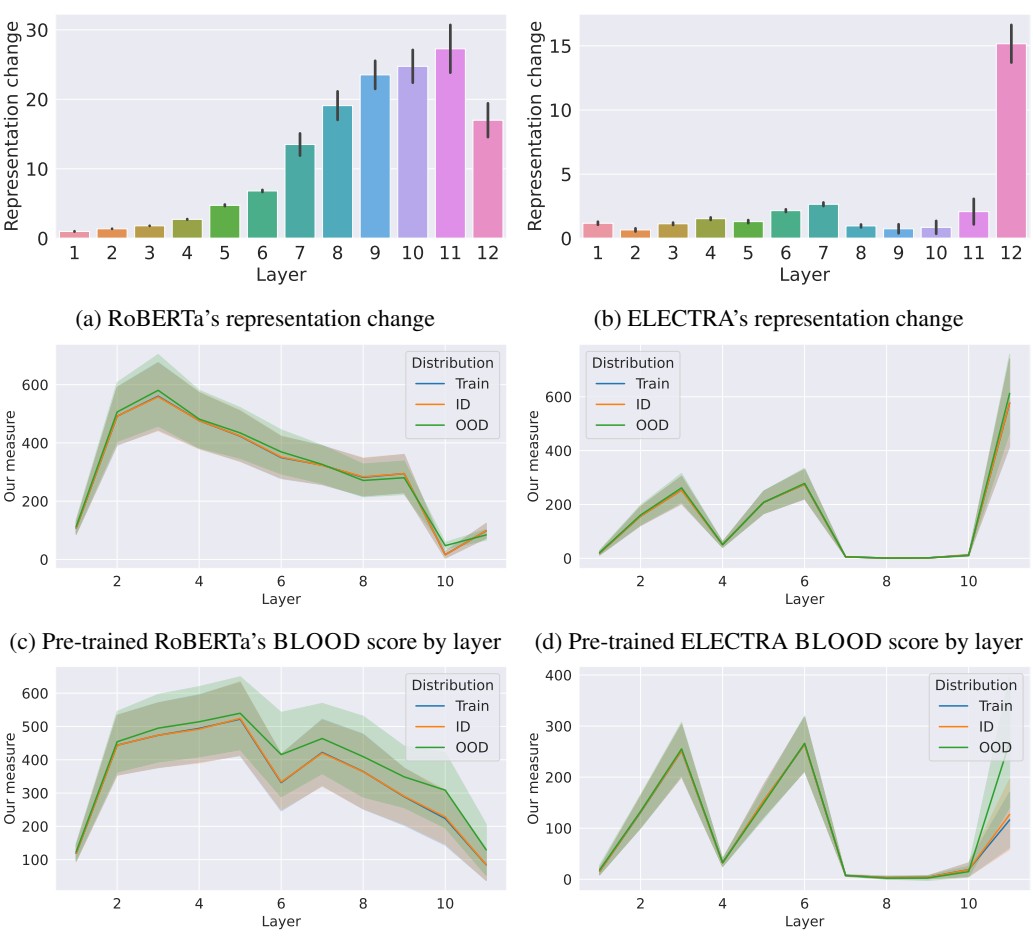

(a) RoBERTa's representation change

(b) ELECTRA's representation change

(c) Pre-trained RoBERTa's BLOOD score by layer

(d) Pre-trained ELECTRA BLOOD score by layer

(e) Fine-tuned RoBERTa's BLOOD score by layer

(f) Fine-tuned ELECTRA BLOOD score by layer

Figure 6: The impact of change of each layer on BLOOD score across layers. Top row: Change in intermediate representations of training instances by layer for (a) RoBERTa and (b) ELECTRA. The scores are averaged across instances for the AGN dataset. The black error bars denote the standard deviation. Middle row: BLOOD score by layer of models for AGN before fine-tuning. Bottom row: BLOOD score by layer of models for AGN after fine-tuning.

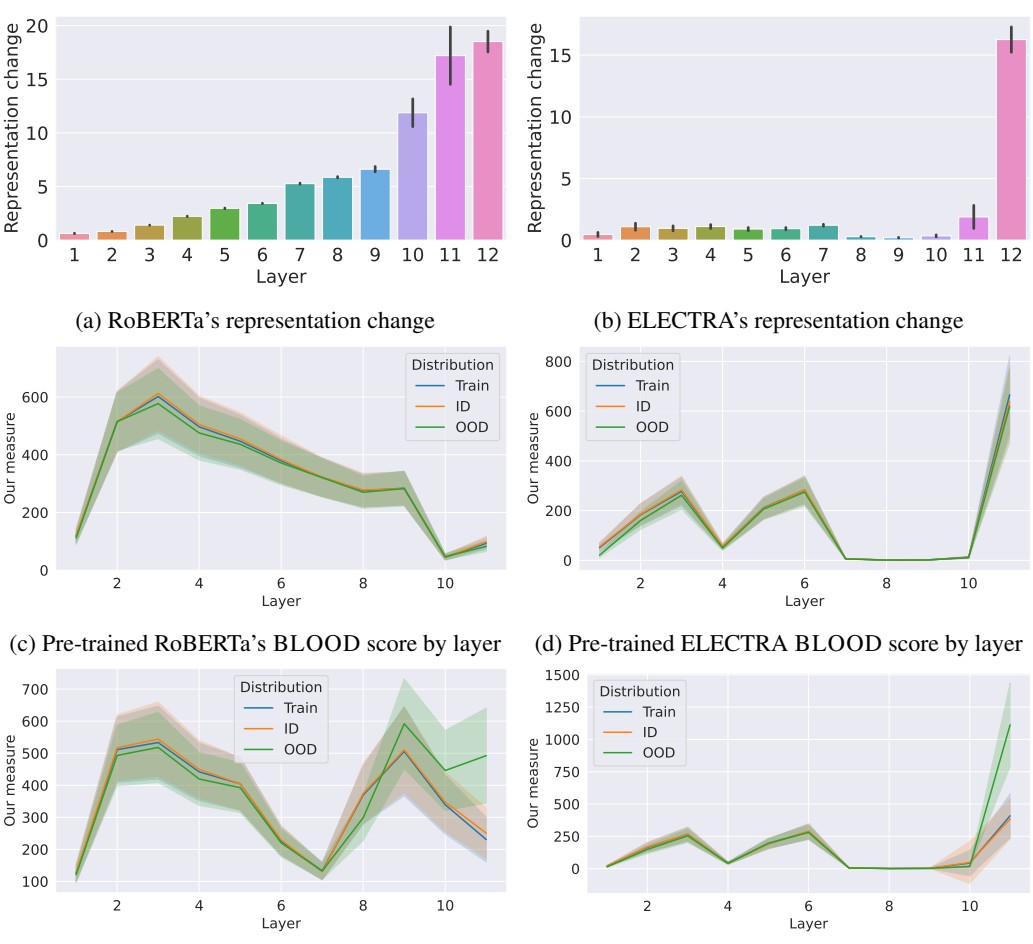

(a) RoBERTa's representation change

(b) ELECTRA's representation change

(c) Pre-trained RoBERTa's BLOOD score by layer

(d) Pre-trained ELECTRA BLOOD score by layer

(e) Fine-tuned RoBERTa's BLOOD score by layer

(f) Fine-tuned ELECTRA BLOOD score by layer

Figure 7: The impact of change of each layer on BLOOD score across layers. Top row: Change in intermediate representations of training instances by layer for (a) RoBERTa and (b) ELECTRA. The scores are averaged across instances for the TREC dataset. The black error bars denote the standard deviation. Middle row: BLOOD score by layer of models for TREC before fine-tuning. Bottom row: BLOOD score by layer of models for TREC after fine-tuning.

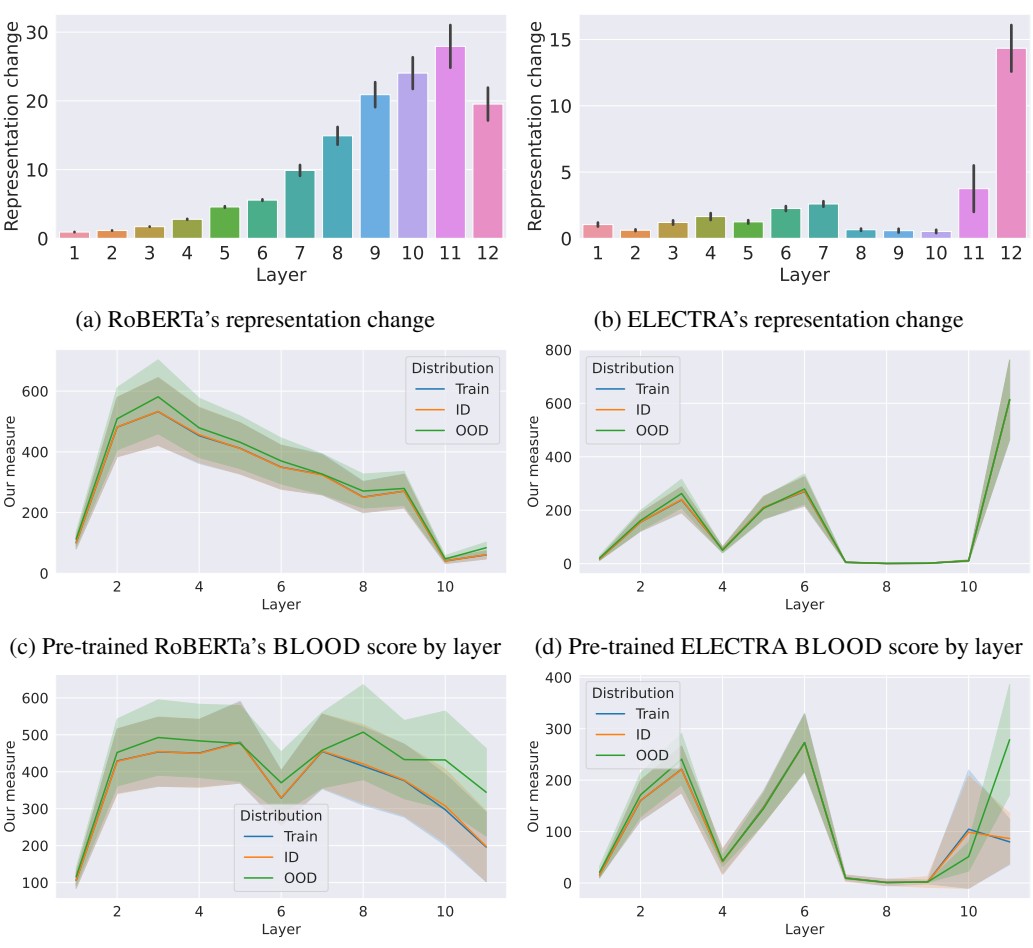

(a) RoBERTa's representation change

(b) ELECTRA's representation change

(c) Pre-trained RoBERTa's BLOOD score by layer

(d) Pre-trained ELECTRA BLOOD score by layer

(e) Fine-tuned RoBERTa's BLOOD score by layer

(f) Fine-tuned ELECTRA BLOOD score by layer

Figure 8: The impact of change of each layer on BLOOD score across layers. Top row: Change in intermediate representations of training instances by layer for (a) RoBERTa and (b) ELECTRA. The scores are averaged across instances for the BP dataset. The black error bars denote the standard deviation. Middle row: BLOOD score by layer of models for BP before fine-tuning. Bottom row: BLOOD score by layer of models for BP after fine-tuning.

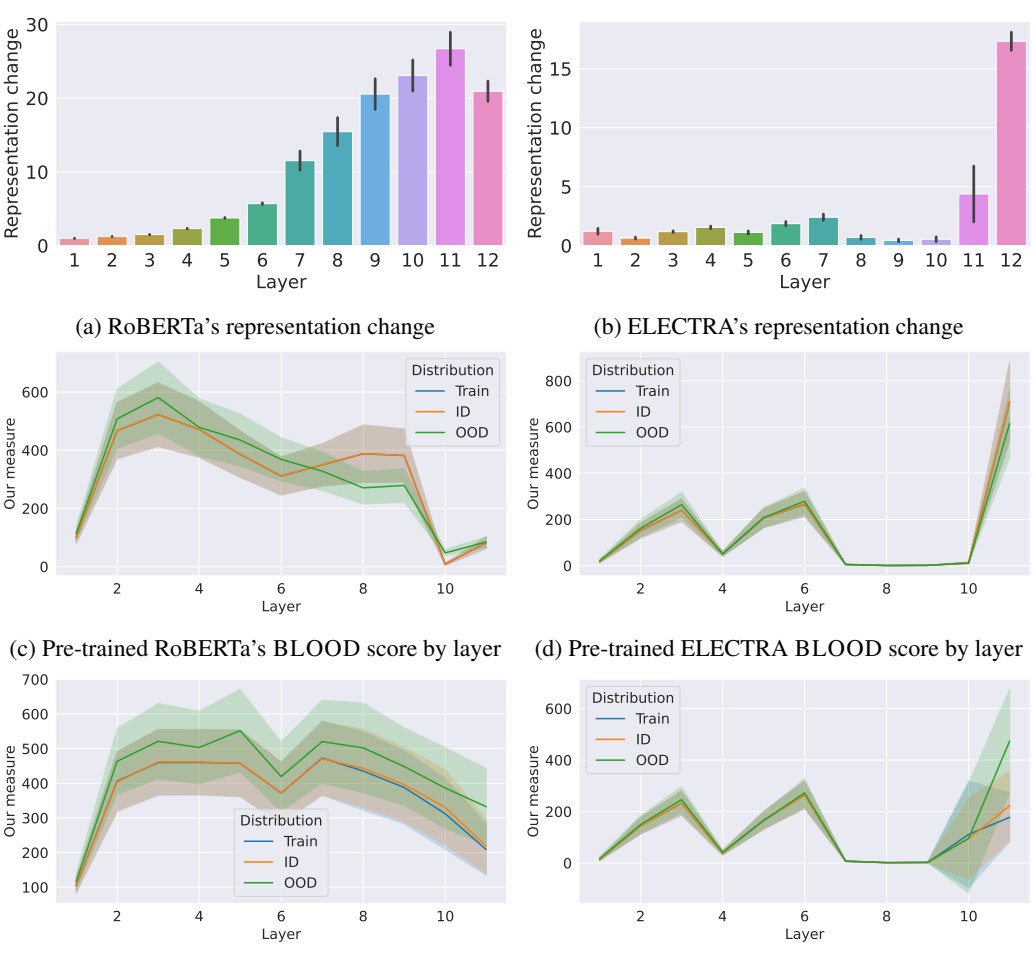

(a) RoBERTa's representation change

(b) ELECTRA's representation change

(c) Pre-trained RoBERTa's BLOOD score by layer

(d) Pre-trained ELECTRA BLOOD score by layer

(e) Fine-tuned RoBERTa's BLOOD score by layer

(f) Fine-tuned ELECTRA BLOOD score by layer

Figure 9: The impact of change of each layer on BLOOD score across layers. Top row: Change in intermediate representations of training instances by layer for (a) RoBERTa and (b) ELECTRA. The scores are averaged across instances for the MG dataset. The black error bars denote the standard deviation. Middle row: BLOOD score by layer of models for MG before fine-tuning. Bottom row: BLOOD score by layer of models for MG after fine-tuning.

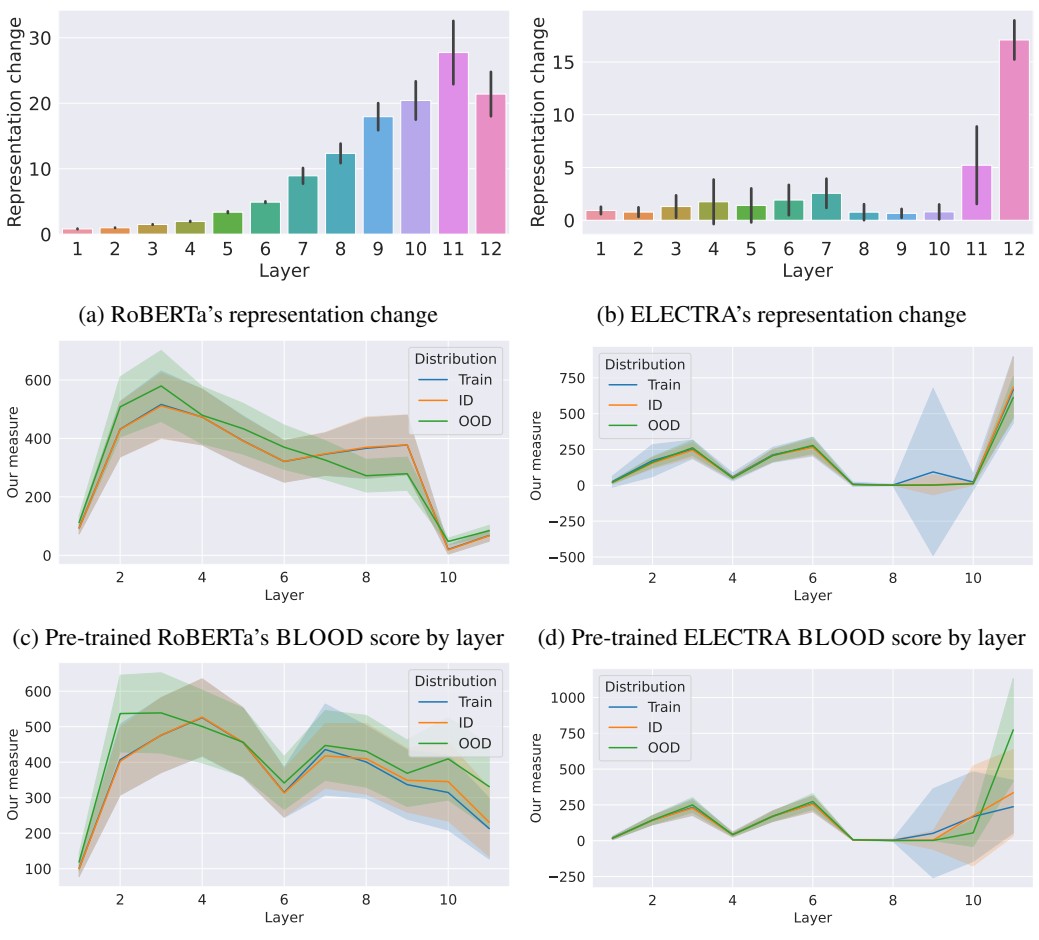

(a) RoBERTa's representation change

(b) ELECTRA's representation change

(c) Pre-trained RoBERTa's BLOOD score by layer

(d) Pre-trained ELECTRA BLOOD score by layer

(e) Fine-tuned RoBERTa's BLOOD score by layer

(f) Fine-tuned ELECTRA BLOOD score by layer

Figure 10: The impact of change of each layer on BLOOD score across layers. Top row: Change in intermediate representations of training instances by layer for (a) RoBERTa and (b) ELECTRA. The scores are averaged across instances for the NG dataset. The black error bars denote the standard deviation. Middle row: BLOOD score by layer of models for NG before fine-tuning. Bottom row: BLOOD score by layer of models for NG after fine-tuning.

