# OpenReview forum: "Out-of-Distribution Detection by Leveraging Between-Layer Transformation Smoothness"
_ICLR.cc/2024/Conference — ICLR 2024 poster_

### Official Review · Reviewer_ELpL · 2023-10-30

**Soundness:** 3 good
**Presentation:** 3 good
**Contribution:** 3 good
**Rating:** 6
**Confidence:** 2

**Summary:**

This paper proposed a white-box OOD detection method: BLOOD, by using the fact that the tendency of between-layer representation transformations of ID data is smoother than the corresponding transformations of OOD data for Transformer network.

-Hypothesis: During the model’s training, smooth transformations between DNN layers are learned, which corresponds to natural and meaningful progressions between abstractions for ID data. And these progressions will not match OOD data, so the transformations will not be smooth.

-The smoothness is defined as the difference of the mapping between the current representation its infinitesimally close neighbourhood's representation.

**Strengths:**

1. Looking at the smoothness of the layer is novel and hasn't been explored in prior research for OOD detection.

2. An unbiased estimator for the smoothness is proposed to reduce the computation.

3. BLOOD demonstrates superior performance over various OOD detection methods for RoBERTa and ELECTRA in text classification datasets.

**Weaknesses:**

1. The experiments exclusively focus on Transformer-based models and text classification. I am curious about the performance of BLOOD when applied to image classification datasets and Convolutional Neural Networks (CNNs). Additionally, the study lacks a comparison with the latest state-of-the-art methods for out-of-distribution (OOD) detection in CNNs, such as ASH (Extremely Simple Activation Shaping for Out-of-Distribution Detection, ICLR23), ReAct (ReAct: Out-of-distribution Detection With Rectified Activations, NeurIPS21), and DICE (DICE: Leveraging Sparsification for Out-of-Distribution Detection, ECCV2022). Furthermore, there is no analysis provided to explain why the concept of BLOOD is restricted to text classification and not applicable to other domains.

2. While Figure 1 qualitatively illustrates the smoothness difference, the author did not offer an explanation for the observed difference between in-distribution (ID) and out-of-distribution (OOD) data. I wonder whether there is an analysis or understanding of this phenomenon, specifically why the learned progressions on ID data do not align with OOD data and how to define this misalignment.

**Questions:**

Please see the weaknesses section, I am willing to raise my rating if the author can addresses those issues.

---

> ### Author Response · Authors · 2023-11-19
> **Response to Reviewer 4**
>
> Thank you for your constructive and helpful comments. We are glad you took notice of the novelty of our work and the usefulness of the estimator we proposed. We will be addressing your comments below in the order they appear in the review:
>
> **1) CV experiments** - Please see our General Response and Updates to the Paper.
>
> **2) More SOTA baselines** - Please see our General Response and Updates to the Paper.
>
> **3) Explanation of difference in smoothness** - On the operational level, BLOOD works because the learning algorithm learns the smooth transitions for the data it was trained on, while the rest of the representation space is either left largely unchanged (if the task was simple enough) or sharpened as the byproduct of trying to fit the more complex data, as we showed in the paper.
> We introduced the intuition and motivation for our method in subsection 3.2. Our intuition is based on the phenomenon known from the literature that different Transformer layers model features at different levels of abstraction. Since the model learns on the ID training data, it should learn the natural progression between the layers of abstractions inherent to the task, i.e., ID data. However, the unseen OOD data will likely have different features in a given level of abstraction than the ID data since the nature of the data is different, so the model is not familiar with them or their progression. Because the model has not learned how to handle the mismatch in the transition between the levels of abstraction, it exhibits sharper between-layer transformations of those representations, i.e., there is a more considerable difference between how OOD representations get mapped onto the next layer and how its infinitesimally small neighborhood gets mapped.
>
> We hope you find our response helpful. If there are any other questions you would like to ask, we would be glad to answer them.

---

> > ### Comment · Reviewer_ELpL · 2023-11-23
> >
> > Thanks for your rebuttal. It addresses some of my concerns. I have revised my score to 6. In the additional experiments, I notice that BLOOD loses its effectiveness when combined with CNNs, but preserves its effectiveness on Transformer.  It would be good if the authors can provide some analysis on discrepancies in the smoothness metric for CNNs and Transformers.

---

### Official Review · Reviewer_woCv · 2023-10-30

**Soundness:** 2 fair
**Presentation:** 3 good
**Contribution:** 3 good
**Rating:** 6
**Confidence:** 4

**Summary:**

This paper proposes a new OOD detection method called BLOOD, designed for a white-box setting where only pre-trained weights and model architectures are accessible, but without access to training data. It leverages the smoothness of between-layer representation transformations in ID data compared to OOD data. The method is evaluated using datasets for text classification tasks, demonstrating its superior performance over other methods with comparable resource requirements.

**Strengths:**

The authors propose a novel and effective scoring method for OOD detection based on the smoothness between Transformer layers that is applicable to pretrained models without access to training data. They provide a detailed performance analysis, including both strengths and limitations. The paper also presents an interesting finding regarding the relationship between smoothness and the complexity of the training dataset. The paper is well-organized and clearly written overall.

**Weaknesses:**

The proposed method does not perform competitively on simpler datasets, such as those with fewer classes in ID, or semantic shift as OODs. This inconsistency in performance across different datasets and settings suggests that relying solely on the smoothness of the representation may not be fully optimal for general OOD detection tasks.

In Table 2, the calculated effect size is greater in the “Mean” approach than in the “Last” approach, which appears to contradict the results in Table 1.

**Questions:**

- In Section 4.3, there are certain inconsistencies in the presented results (e.g., RoBERTa model on the MG and NG datasets), which seem to challenge the hypotheses by the authors. Given the variations in performance, can you provide additional evidence or discussion, such as whether certain architectures or settings are preferred by the proposed method?
- I’m curious if BLOOD can be extended to open-box settings.
- Is it possible to apply BLOOD to OOD detection tasks in computer vision? Any insights or preliminary findings on this would be valuable.
- Could you provide more detailed explanation of how the CLES measurements in Tables 2 and 3 were computed?

---

> ### Author Response · Authors · 2023-11-19
> **Response to Reviewer 3 (1/2)**
>
> Thank you for showing interest in our paper with many thought-provoking questions. We are glad you enjoyed the thorough analysis of our method and our findings on smoothness in DNNs. We respond to your questions and comments in order of how they appear in the review:
>
> **1) There are scenarios in which BLOOD is outperformed by other OOD methods** - It is true that in certain scenarios BLOOD is outperformed by other popular OOD detection methods, e.g., for simpler training datasets or semantic shifts. However, the performance boosts in the scenarios suitable for BLOOD are significant (e.g., 12.09 percentage points AUROC better than the second-best white-box method for BigPatent dataset with ELECTRA). That is precisely why we carried out an extensive analysis of when BLOOD outperforms other methods and why it outperforms them when it does. Researchers and engineers must be able to make an educated choice when deciding on the suitable OOD detection method for their problem, and that is why we believe it is essential to showcase both the strengths and limitations of our work.
>
> **2) Table 2 contradicts Tabe 1** - No, that is as expected. A lower CLES implies a higher AUROC because the learning algorithm has sharpened the OOD region of representation space along with the smoothening of the ID region instead of just smoothening the ID region. That emphasizes the difference in smoothness between ID and OOD data. And since the learning algorithm had to “invest more work” in the last layers (as seen in Table 1), it sharpened the OOD region of representation space as a byproduct, leading to lower CLES for the last layer and higher AUROC when using the last layer.
> The reasoning behind the CLES of lower vs. higher layers is parallel to the reasoning behind the CLES of simpler vs. more complex datasets. If a dataset is more complex, the learning algorithm changes the model more during training, which, instead of just focusing on the ID region of the representation space, also makes changes to the OOD region of the representation space as a byproduct. Similarly, in higher layers, which are more influenced by the learning algorithm, there are more changes in the OOD region of space.
>
> Because of that, the only instances where CLES in the last layer is larger than the CLES_mean for RoBERTa is for theMG and NG datasets, which are exactly the datasets on which BLOOD_{mean} outperforms BLOOD_{L-1} on RoBERTa.
>
> **3)  Relationship between BLOOD_mean and BLOOD_{L-1}** - Please see our General Response and Updates to the Paper.
>
> **4) BLOOD in an open-box setting** - While we focused on the white-box scenarios because of the wider useability of our method, we are thankful for this great suggestion.
> We experimented with extending BLOOD by using a small sample of OOD examples to learn the optimal parameters for OOD detection (as is done by many popular OOD methods), e.g., learn the optimal weights for the weighted average of scores in different layers. We used a validation set of 5% of original training sets and fitted a logistic regression to learn the optima weights for a weighted sum of BLOOD values in different layers. This approach generally outperforms the open-box versions of our method. We provide results (AUROC) of this experiment below. We will be adding the results in the Appendix of our paper:
>
>
> | Model   | SST   | SUBJ  | AGN   | TREC  | BP    | AR    | MG    | NG    |
> |---------|-------|-------|-------|-------|-------|-------|-------|-------|
> | RoBERTa | 73.94 | 82.98 | 81.39 | 91.73 | 93.47 | 96.25 | 92.46 | 89.97 |
> | ELECTRA | 77.67 | 77.30 | 82.76 | 98.73 | 96.54 | 91.97 | 90.92 | 85.19 |

---

> > ### Author Response · Authors · 2023-11-19
> > **Response to Reviewer 3 (2/2)**
> >
> > **5) CV experiments** - Please see our General Response and Updates to the Paper.
> >
> > **6) Detailed explanation of CLES** - We take the intermediate representations of the ID and OOD data in each of the layers $l$ denoted by $r^l_P$ from the pre-trained model (i.e., before fine-tuning on training data). Then, after we trained the model on training data, we again took intermediate representations of ID and OOD data from each layer denoted by $r^l_F$. We then calculate the Euclidean distance between representations from the model before fine-tuning and the representations from the model after fine-tuning $c^l = || r^l_P-r^l_F|| $. We then use this measure $c^{l}$, which represents how much a particular data point’s intermediate representation has changed after training in the layer $l$, and calculate two different quantities: (1) MEAN = $1/L \sum_l c^l$ and (2) LAST = $c^L$, where $L$ is the last layer.
> >
> > We then order all ID and OOD points based on their value of either $c_{MEAN}$ or $c_{L}$. To calculate the CLES, we create a Cartesian product between ID and OOD data points and then calculate the percentage of times ID data points have larger $c$ than OOD data points (it is mathematically equivalent to taking the AUROC of ID and OOD data based on $c$).
> > The interpretation of CLES is that (1) if CLES is larger than 50% ID representations changed more than OOD representations, (2) if CLES is close to 50% ID and OOD representations changed roughly the same, and (3) if CLES is less than 50% ID representations changed less than OOD representations.
> >
> >
> > We hope you find our response helpful. If you have any other questions or suggestions, please let us know. We would be happy to discuss them further.

---

### Official Review · Reviewer_Bnog · 2023-10-31

**Soundness:** 2 fair
**Presentation:** 3 good
**Contribution:** 2 fair
**Rating:** 5
**Confidence:** 4

**Summary:**

The paper introduces an out of distribution detection method called BLOOD which measures the transformation smoothness between layers by using the frobenius norm on the jacobian matrix. The proposed approach is then evaluated across several text classification dataset for OOD detection task.

**Strengths:**

- The proposed method is simple and can be easily applied on different tasks and network architectures, where OOD detection is required.
- The paper is easy to read and overall the paper is well written.
- The proposed method has been shown to perform well across different text classification datasets in comparison to standard OOD methods.

**Weaknesses:**

- The experimental results are not very strong or convincing. I would have expected some experiments on vision datasets as there has been a significant amount of OOD detection work focussing on vision datasets. Also, this approach can be easily applied on those tasks and will give better understanding how well this approach works.
- Most of the baselines considered in the comparisons are very old for OOD detection. There has been a significant amount of work focusing on last layer contributions to OOD detection similar to this paper (for eg. [a,b]). The authors should show comparisons against SOTA OOD baselines.



References:
- [a] Sun et. al. React: Out-of-distribution detection with rectified activations. Neurips, 2021.
- [b] Djurisic et al. Extremely simple activation shaping for out-of-distribution detection. arXiv preprint arXiv:2209.09858, 2022

**Questions:**

Please address the weakness mentioned above.

Why does BLOOD_mean performance significantly worse than the baselines in most of the cases? Whereas in some case it performs better than BLOOD_{L-1}. Can the authors explain that?

---

> ### Author Response · Authors · 2023-11-19
> **Response to Reviewer 2**
>
> Thank you for your time and effort in reviewing our paper. We appreciate you finding our paper well-written and our results strong. All other reviewers shared your suggestions, so we acknowledged them in our General Response and Updates to the paper. We provide the summary of our answers below, but please see General Response for more detailed answers:
>
> **1) CV experiments** - BLOOD is competitive and all of the conclusions of our paper hold when applied to vision datasets paired with Vision Transformer. However, when paired with CNNs, BLOOD behaves differently, likely due to differences in how the two architectures process data.
>
> **2) More SOTA baselines** - We expanded our experiments with the suggested methods (ASH and ReAct), however those methods did not retain their SOTA status on the text classification task.
>
> **3) Relationship between BLOOD_mean and BLOOD_{L-1}** - BLOOD_mean introduces noise to the method because it includes BLOOD scores from the lower layer that did not change much during training and thus retain their BLOOD scores from the pre-trained models where there is no difference between ID and OOD data. And since BLOOD scores in the low layers have higher magnitudes, they take over the average. However, for datasets MG and NG on RoBERTa, BLOOD_mean works better than BLOOD_{L-1}. The reasoning behind that is the same (BLOOD scores are kept the same as before training in low layers), but in these scenarios, there exists a priori difference in BLOOD score between ID and OOD data, as seen from Figures 9 and 10.
>
> We hope you find our answers helpful. If you have any other questions, we would be happy to hear them.

---

### Official Review · Reviewer_Txyj · 2023-11-01

**Soundness:** 3 good
**Presentation:** 3 good
**Contribution:** 3 good
**Rating:** 5
**Confidence:** 3

**Summary:**

This paper tackles OOD detection problems by analyzing the smoothness of the feature transformation between intermediate layers in a network in a pre-trained network. The idea behind the notion of smoothness in this paper extends from Liptischitz continuity. The method is evaluated on popular pre-trained models such as RoBERTa and ELECTRA on multiple texts analysing data sets such as SST, SUBJ, AGN etc. The results are compared with competitive baselines, which demonstrates its effectiveness in most of the cases.

**Strengths:**

OOD detection is an important research problem that plays a significant role in developing trustworthy AI. Hence, the paper addresses the problem which can be of interest to a larger audience.

The method itself relies on the transformation features between the layers, which are estimated from the pre-trained models. Hence, this method does not require labelled examples from downstream tasks which is another strength of this method.

**Weaknesses:**

In the paper, it is mentioned that the method is robust to complex tasks and less competitive for easy tasks. However, the explanations are just based on empirical performance, lacking clear insight and understanding behind such outcomes.

This is a similar line of work on OOD detection employing variance of gradients [a], which play a direct role in the smoothness feature transformation in the intermediate roles.
It is better this paper acknowledges such works and argues how such methods differ from their proposed method.

[a] Agarwal, Chirag, Daniel D'souza, and Sara Hooker. "Estimating example difficulty using variance of gradients." Proceedings of the IEEE/CVF Conference on Computer Vision and Pattern Recognition. 2022.

**Questions:**

Please see the weakness.

I wonder how this method would work in vision tasks.

---

> ### Author Response · Authors · 2023-11-19
> **Response to Reviewer 1**
>
> Thank you for your constructive comments and questions, we are glad you find our work important. We address your questions below, as introduced in your review:
>
> **1) Insights on why BLOOD works better on complex datasets** - In subsection 4.4, we test the hypothesis that BLOOD works better on simpler datasets by simplifying the more complex datasets. In that section, we show that reducing the dataset's complexity leads to an increase in CLES of the change in representations. This implies that the model can fit the data by only focusing on the parts of the representation spaces populated by the ID training data. However, for more complex datasets, the model has to undergo more radical changes during training to fit the data, sharpening the OOD part of the representation space as the byproduct. Because BLOOD compares the smoothness of the ID region and OOD region of the representation space, sharpening the OOD region of the space, along with smoothening the ID region of the space, emphasizes the difference in smoothness.
>
> **2) Acknowledge the "Estimating example difficulty using variance of gradients (VoG)"** - Thank you for drawing our attention to this paper. It is an exciting work with slight similarities to our work. However, there are also crucial differences.
> One conceptual difference is that VoG considers the gradient of the inputs w.r.t. the logit of the true/predicted class (measuring how much the model's predictions vary with small changes in the input), while BLOOD is analyzing the Jacobians of neighboring intermediate layers (quantifying the stability of the intermediate representations during inference).
> Another practical difference is that VoG calculates the variance of the gradients – for which it needs the gradients at different timesteps during training. VoG requiring access to the model's training process makes it an open-box method, while BLOOD is a post-hoc white-box method, allowing it to be more widely used.
>
> **3) CV experiments** - Please see our General Response and Updates to the Paper.
>
> We hope we have addressed all of your questions. If there is anything else you would like to know about our work, please do not hesitate to ask.

---

### Author Response · Authors · 2023-11-19
**General Response (1/2)**

Firstly, we thank the reviewers* for their time and constructive comments that will help us improve our paper. We are glad that reviewers find our method **effective and easy to use** (R1, R2, R3, R4), our paper **well-written** (R2, R3), and our research problem **significant and potentially attractive to a larger audience** (R1).  We are grateful that reviewers appreciated the **novelty** of our method since we present an **original approach** that does not extend any prior work (R3, R4). We are also pleased that reviewers recognized the **usefulness of the estimator we derived** for efficiently computing the norm of the Jacobian in deep neural networks (R4), which can potentially have a much wider use beyond OOD detection. Finally, we are encouraged by recognizing our **thorough analysis of both the strengths and limitations** of our approach (R3) and an appreciation for our **findings that improve the understanding of smoothness in deep neural networks** (R3).

Here, we will share our response to the comments shared by multiple reviewers. But first, we would like to acknowledge the slight mistake in the submitted paper. In Table 1, we tested the statistical significance of our result using the **one-sided** test, but we should have stated that explicitly. We will clarify the test type used in the paper's final version.

**1) More SOTA baselines** - A common suggestion (R2, R4) was to compare our method with more SOTA methods. This may strengthen our findings, and we expand our experiments with two methods suggested by both reviewers: ASH and ReAct. For the experiments, we used ASH-S with $p=90%$ and ReAct with $p=90%$ (as suggested by the original papers), both applied to the activations of the penultimate layer. We provide the results of the experiments on the image classification task here and in the updated paper:


| Model   | Method | SST   | SUBJ  | AGN   | TREC  | BP    | AR    | MG    | NG    |
|---------|--------|-------|-------|-------|-------|-------|-------|-------|-------|
| RoBERTa | ReAct  | 69.55 | 73.33 | 77.10 | 96.05 | 86.19 | 92.65 | 87.30 | 83.17 |
| RoBERTa | ASH    | 67.22 | 79.27 | 72.54 | 90.36 | 82.18 | 91.42 | 81.62 | 77.73 |
| ELECTRA | ReAct  | 71.18 | 68.33 | 79.46 | 97.50 | 85.26 | 91.01 | 81.22 | 80.95 |
| ELECTRA | ASH    | 67.92 | 75.11 | 77.96 | 90.18 | 79.81 | 83.96 | 71.84 | 74.50 |


Similarly to [1], our results show that the proposed methods do not retain their SOTA performance on text classification tasks. It seems that distance based methods (e.g., KNN and Mahalanobis distance) perform best on text classification. For more information, please see Table 1 in the revised paper.


**2) CV experiments** - A critique shared by all of the reviewers (R1, R2, R3, R4) is that we conducted the experiments only on the text classification datasets and not on any computer vision tasks. We did that because we mainly work on NLP, but we understand the interest in the evaluation of our method on CV tasks and models. We appreciate the suggestion and, since there is nothing that prevents our method of working with image classification, we agree with the authors that the inclusion of those experiments would significantly improve our paper. We provide the preliminary results of the experiments on the image classification task here and in the updated version:


| Model    | ID       | OOD   | BLOOD_M   | BLOOD_L   | MSP   | ENT       | EGY       | GRAD  |
|----------|----------|-------|-----------|-----------|-------|-----------|-----------|-------|
| ResNet34 | cifar10  | Beans | 85.68     | 14.16     | 89.23 | 91.55     | **92.41** | 81.04 |
| ResNet34 | cifar100 | Beans | **86.40** | 23.54     | 73.32 | 79.90     | 81.22     | 76.55 |
| ResNet50 | cifar10  | Beans | 78.37     | 50.28     | 79.18 | **80.45** | 79.02     | 69.33 |
| ResNet50 | cifar100 | Beans | **86.40** | 49.76     | 79.13 | 80.79     | 82.31     | 85.29 |
| ViT      | cifar10  | Beans | 95.41     | 99.58     | 99.31 | 99.31     | **99.98** | 96.92 |
| ViT      | cifar100 | Beans | 91.68     | **96.53** | 96.02 | 95.71     | 95.81     | 89.33 |
| ResNet34 | cifar10  | SVHN  | 84.00     | 49.99     | 88.92 | **89.71** | 88.23     | 82.29 |
| ResNet34 | cifar100 | SVHN  | **87.05** | 49.51     | 73.64 | 76.20     | 77.90     | 85.97 |
| ResNet50 | cifar10  | SVHN  | 84.13     | 49.91     | 89.55 | **90.29** | 84.36     | 88.01 |
| ResNet50 | cifar100 | SVHN  | 79.77     | 50.50     | 79.86 | **83.32** | 82.70     | 81.80 |
| ViT      | cifar10  | SVHN  | 99.37     | 92.45     | 99.12 | 98.19     | **99.55** | 96.94 |
| ViT      | cifar100 | SVHN  | 80.97     | **89.07** | 84.06 | 84.58     | 88.21     | 85.53 |

---

> ### Author Response · Authors · 2023-11-19
> **General Response (2/2)**
>
> It can be seen from the results that our conclusions are retained when BLOOD is applied to Vision Transformer (ViT), BLOOD_{L-1} works better than BLOOD_mean and BLOOD performs better on more complex cifar100 than on simpler dataset cifar10. However, it seems that BLOOD does not retain its effectiveness when paired with CNNs, i.e., ResNet34 and ResNet50. Also, for CNNs, BLOOD_mean works fine for detecting OOD data, while BLOOD_{L-1} is not discriminative between OOD and ID data. For more details, please see Appendix D of our updated paper.
>
> There are inherent differences in how different architectures process data, which impact the availability of different OOD detection methods. In our experiments and the experiments in [1], the best performing OOD methods for Transformer-based language models are distance-based methods (e.g., KNN and Mahalanobis distance), while methods that are SOTA on CNNs (ReAct, ASH, DICE) often underperform even against the simple baselines. In [1], DICE is by far the worst-performing OOD detection method for Transformer-based LLMs. Conversely, in the original paper of ASH [2], the authors did a comprehensive OOD detection experiment on CNNs, which showed Mahalanobis distance by far underperforming compared to other methods. In light of the finding that the choice of architecture plays a crucial role in the success of the OOD detection method, we updated our paper to reflect that Transformers are the scope of our work.
>
> **3) Relationship between BLOOD_mean and BLOOD_{L-1}** - Reviewers R2 and R3 were interested in the reasoning behind BLOOD-mean being the worst choice most of the time, except in a few instances. Firstly, let us explain why BLOOD_mean underperforms (especially for RoBERTa). Figure 1 and similar figures in the Appendix for other datasets (Figures 4-10) show how BLOOD changes across layers. Comparing the subfigures it can be seen that since there is almost no change after training in the lower layers, the BLOOD in those layers stays the same as before training, where there was no difference between the ID and OOD data (because the model has not yet seen the ID data). However, in those lower layers for RoBERTa the BLOOD scores have larger magnitudes than in higher layers where the difference in BLOOD of ID and OOD data is prominent. Thus, by averaging the BLOOD scores of different layers the noisy lower layers dominate the informative higher layer and make BLOOD less discriminative between ID and OOD data. Since for ELECTRA the higher and more informative layers have higher magnitudes of BLOOD score, this effect is less noticeable.
> However, the reviewers have noticed that in some cases, e.g., on MG and NG datasets with RoBERTa, BLOOD_mean archives better results. We can look at Figures 9 and 10 to explain this effect. Looking at subfigures (c) it can be seen that there is a difference in BLOOD scores between ID and OOD data across layers even before fine-tuning on the ID data, likely due to inherent differences in ID and OOD data (we speculate it could be due to the length of documents, as the longest documents from our experiments come from MG and NG datasets). It can be seen that in low layers OOD data has higher BLOOD scores than ID data, and that difference is retained even after fine-tuning, as those layers are not changed much. Thus, including the lower layers helps discriminate between ID and OOD data.
> However, we would like to note that it is still better and more consistent to use BLOOD_{L-1} than BLOOD_mean, since this difference in the earlier layers could have gone the other way around and impact the performance of BLOOD_mean in a negative way.
>
> We have addressed all other comments in individual responses to each reviewer. The reviews have significantly improved our paper, for which we are thankful.
>
>
> *As abbreviations, we refer to reviewers **Txyj** as R1, **Bnog** as R2, **woCv** as R3, and **ELpL** as R4 respectively.
>
> [1] Mateusz Baran, Joanna Baran, Mateusz Wójcik, Maciej Zięba, and Adam Gonczarek. 2023. Classical Out-of-Distribution Detection Methods Benchmark in Text Classification Tasks. In Proceedings of the 61st Annual Meeting of the Association for Computational Linguistics
>
> [2] Andrija Djurisic, Nebojsa Bozanic, Arjun Ashok, and Rosanne Liu. Extremely simple activation shaping for out-of-distribution detection. In The Eleventh International Conference on Learning Representations, ICLR 2023

---

### Author Response · Authors · 2023-11-19
**Updates to the Paper**

We thank the reviewers again for strengthening our paper with their suggestions. We updated our original paper with the reviewers’ suggestions. We summarize the changes below:

- **CV experiments** (R1, R2, R3, R4) - Appendix D
- **Comparison to ASH and ReAct** (R2, R3) - Tables 1, 3, and 4
- **Acknowledge ASH and VoG in RW** (R1) - 2. Related Work
- **BLOOD as an open-box method** (R3) - Appendix E
- **Fix the caption of Table 1** - Clarification that the Man-Whitney U test was one-sided
- **Fix Figure 4 in the Appendix** - subfigures (c) and (e) were accidentally the same
- **Narrowed focus to Transformers** - Abstract, Introduction, Intuition, Conclusion

---

### Meta-Review · Area_Chair_5Hry · 2023-12-10

**Metareview:**

The paper introduces an out of distribution detection method called BLOOD, which analyzes the smoothness of the feature transformation between intermediate layers in a pretrained network, by using the frobenius norm on the jacobian matrix.

The method is evaluated on popular pre-trained models such as RoBERTa and ELECTRA on multiple texts classification datasets. The results are compared with competitive baselines, which demonstrates its effectiveness in most of the cases. During rebuttal, the author added computer vision tasks, although the improvement there is unclear.

Overall, It is a novel and interesting method with promising results on natural language tasks.

**Justification For Why Not Higher Score:**

Method only seems to work on text datasets. It's not a flaw of itself, but it would be great for the author to keep exploring and understanding why the difference of performances across tasks.

**Justification For Why Not Lower Score:**

The method was clearly stated and theoretically backed, the results on text classifcation are clear and strong.

---

### Decision · Program_Chairs · 2024-01-16

Accept (poster)